# A satellite-based estimate of combustion aerosol cloud microphysical effects over the Arctic Ocean

Lauren M. Zamora[1,2], Ralph A. Kahn[2], Klaus B. Huebert[3], Andreas Stohl[4], Sabine Eckhardt[4]

[1]Earth System Science Interdisciplinary Center, University of Maryland, College Park, MD, USA
[2]NASA Goddard Space Flight Center, Greenbelt, MD, USA
[3]Horn Point Laboratory, University of Maryland Center for Environmental Science, Cambridge, MD, USA
[4]NILU – Norwegian Institute for Air Research, Kjeller, Norway

*Correspondence to*: Lauren M. Zamora (lauren.m.zamora@nasa.gov)

**Abstract.** Climate predictions for the rapidly changing Arctic are highly uncertain, largely due to a poor understanding of the
processes driving cloud properties. In particular, cloud fraction (CF) and cloud phase (CP) have major impacts on energy budgets, but are poorly represented in most models, often because of uncertainties in aerosol-cloud interactions. Here we use over 10 million satellite observations coupled with aerosol transport model simulations to quantify large-scale microphysical effects of aerosols on CF and CP over the Arctic Ocean during polar night, when direct and semi-direct aerosol effects are minimal. Combustion aerosols over sea ice are associated with very large (~10 W m$^{-2}$) differences in longwave cloud radiative
effects at the sea ice surface. However, co-varying meteorological changes on factors such as CF likely explain most of this signal. For example, combustion aerosols explain at most 40% of the CF differences between the full dataset and the clean-condition subset, compared to the 57-91% of the differences that can be predicted by co-varying meteorology. After normalizing for meteorological regime, aerosol microphysical effects have small but significant impacts on CF, CP, and precipitation frequency on an Arctic-wide scale. These effects indicate that dominant aerosol-cloud microphysical mechanisms
are related to the relative fraction of liquid-containing clouds, with implications for a warming Arctic.

## 1 Introduction

Cloud cover has a major influence on surface heating, precipitation, and future climate over the Arctic (Boucher et al., 2013), and may play a role in the enhanced warming over the Arctic compared to lower latitude regions (Södergren et al., 2017), known as Arctic amplification. Aerosols can influence a number of factors relevant to cloud fraction (CF), including cloud
droplet number, phase, lifetime, and probability of precipitation (Albrecht, 1989; Coopman et al., 2016; Girard et al., 2005; Lance et al., 2011; Lindsey and Fromm, 2008; Zamora et al., 2017). However, the regional-scale importance of aerosol microphysical processes on CF has been difficult to constrain from observations and models, particularly due to uncertainties in how aerosols affect precipitation and ice nucleation rates (Gettelman, 2015; Morrison et al., 2012; Ovchinnikov et al., 2014). This is mainly due to the complexity of the responses of different types of clouds to different aerosol types and concentrations
(Fan et al., 2016; Fu and Xue, 2017; Stevens et al., 2017; Zhao et al., 2018), poorly constrained aerosol concentrations

(particularly in winter and beneath thick cloud cover), and confounding effects from co-varying meteorology (Gryspeerdt et al., 2016). These uncertainties contribute to the large uncertainties in model CF and cloud phase (CP) (de Boer et al., 2011; Cesana and Chepfer, 2012; Chernokulsky and Mokhov, 2012; Kay et al., 2010; Liu et al., 2011; Qian et al., 2012; Stanfield et al., 2014; Zib et al., 2012). To account for the impact of meteorological co-variability on Arctic CF, observations covering

large spatial and temporal domains are required, making it difficult to estimate the regional importance of aerosol microphysical effects solely from *in situ* observations. These uncertainties have precluded better constraints on the mechanisms by which aerosols affect cloud microphysics (Morrison et al., 2012) and on general model estimates of their overall importance.

In this paper, we focus on the effects of combustion-derived (i.e., anthropogenic pollution plus smoke) aerosols on clouds.
Combustion-derived aerosols are strongly impacted by anthropogenic activity, and tend to dominate columnar mass under high AOD conditions in the Arctic (Xie et al., 2018), although in spring the more well-mixed mineral dust can also contribute ~10% to total Arctic AOD levels (Breider et al., 2014; Groot Zwaaftink et al., 2016). Combustion aerosols have pronounced effects on Arctic cloud microphysical and radiative properties (e.g., Carrió et al. (2005); Coopman et al. (2016, 2018); Earle et al. (2011); Garrett et al. (2004); Jouan et al. (2014); Lubin and Vogelmann (2006); Tietze et al. (2011); Zamora et al. (2016,
2017); Zhao and Garrett (2015)). Their cloud impacts are likely to be particularly large during winter and spring, when they are transported to the Arctic most efficiently, and when precipitation is reduced, causing a peak in aerosol abundance at many remote Arctic ground stations known as "Arctic Haze" (Barrie, 1986; Croft et al., 2016; Quinn et al., 2007; Stohl, 2006). However, so far it has been challenging to assess their cloud effects on the Arctic region as a whole, due to large cloud model uncertainties, spatial/temporal observation limitations, and difficulties obtaining some types of remote sensing information at
high latitudes.

Here we provide the first observation-based constraint on combustion aerosol microphysical effects on total nighttime CF over the Arctic Ocean region within the spectrum of current-day aerosol and meteorological conditions. Estimates of combustion aerosol microphysical effects on operational, satellite-defined CF are obtained from two years of remote sensing data and
output from the FLEXPART Lagrangian particle dispersion model (Stohl et al., 1998, 2005). We use the model to provide estimates of how clean or polluted the observed air masses were. We then identify average cloud property differences over sea ice and open ocean regions between the full dataset (hereafter referred to as "all conditions") and a subset where combustion aerosols are at clean background levels ("clean conditions"). By comparing clean to all conditions, as opposed to comparing clean to non-clean or polluted-only conditions, our estimates account for the relatively high frequency of clean and low-aerosol
conditions, and are more representative of the microphysical effects of combustion-derived aerosols over sea ice and open ocean regions. It also reduces the need to introduce an additional, arbitrary pollution threshold. By averaging millions of observations after stratifying them by temperature and relative humidity, we minimize confounding effects of local and large-scale meteorological co-variability. A major strength of this method, which depends primarily on direct observations, is that it

requires no detailed parameterization of the fraction of cloud-active aerosols or underlying microphysical mechanisms to constrain the importance of large-scale aerosol microphysical effects on Arctic clouds.

## 2 Methods

Tropospheric cloud data gathered from CloudSat and CALIPSO data during polar night over the Arctic Ocean were collected from 1 January 2008 to 7 December 2009, between 60 and 82° N and 0.6-8.5 km above sea level. These years had typical moisture fluxes and total cloud fractions compared to other recent years (Boisvert and Stroeve, 2015; Kay and L'Ecuyer, 2013). We focused on observations during polar night (solar zenith angle, SZA, > 90°) mainly to isolate indirect effects of aerosols on clouds as much as possible from confounding direct and semi-direct aerosol radiative effects. The nighttime focus also reduces uncertainties from any residual diurnal changes in cloud fraction, and is associated with a better lidar signal-to-noise ratio (used for aerosol transport model validation, see Supplemental Data). Data from all months meeting the above criteria were included except those between May and July. These were excluded to avoid geographic bias in the analysis, as the few nighttime data that were available during this period tended to occur mainly at the lowest latitudes. Clouds below 0.6 km were not assessed due to near-ground uncertainties in the CloudSat and CALIPSO data (de Boer et al., 2009; Liu et al., 2017).

Oceanic areas were determined by ETOPO1 Bedrock GMT4 data (Amante and Eakins, 2009). Oceanic clouds were separated into open ocean and sea ice regions following Zamora et al. (2017): for each profile, the corresponding monthly fractional sea ice cover was determined from the NOAA/NSIDC Climate Data Record of Passive Microwave Sea Ice Concentration, version 2 (Meier et al., 2013; Peng et al., 2013), and samples associated with > 80% or < 20% monthly sea ice fractions were classified as being over sea ice or open ocean, respectively.

### 2.1 Aerosol transport model

Passive remote sensors provide no aerosol data at night, and do not provide reliable aerosol data over bright sea ice or under clouds. Active lidar signals are often attenuated in clouds. Moreover, active sensors such as CALIPSO cannot always detect dilute aerosols, even in conditions with the highest lidar sensitivity (i.e., above clouds at night (Zamora et al., 2017)). Therefore, the presence of combustion aerosols for comparison to the satellite cloud data was estimated with vertically-resolved modeled black carbon (BC) aerosol estimates from the FLEXible PARTicle dispersion model (FLEXPART) (Stohl et al., 1998, 2005). Here, FLEXPART was driven with meteorological analysis data from the European Centre for Medium-Range Weather Forecasts (ECMWF) at a resolution of 1° longitude and 1° latitude. Anthropogenic and biomass burning BC emissions were based on the ECLIPSE (Evaluating the CLimate and Air Quality ImPacts of ShortlivEd Pollutants) (Stohl et al., 2015) and GFED (Global Fire Emission Database) (Giglio et al., 2013) inventories, respectively. Model output was produced in five vertical layers between 0.6-1.5, 1.5-2.5, 2.5-4, 4-6, and 6-8.5 km. Note that the resolution of the meteorological input data is much higher (91 levels) and, as a Lagrangian model, FLEXPART has no discrete resolution for the particle transport. BC

emissions were based on the ECLIPSE emission inventory (Stohl et al., 2015). Note that emission fluxes in the model rely on inventories of emission factor measurements that partially include thermooptical measurements, which may not always completely differentiate between BC and "brown" or organic carbon (BrC or OC) (Russell et al., 2014; Samset et al., 2018). Further details on the model and its configuration can be found in Zamora et al. (2017).

FLEXPART is widely used, and is well-validated for the purpose of studying Arctic smoke and pollution transport (Damoah et al., 2004; Eckhardt et al., 2015; Forster et al., 2001; Paris et al., 2009; Sodemann et al., 2011; Stohl et al., 2002, 2003, 2015; Zamora et al., 2017). FLEXPART BC is used in this study as a proxy for all combustion aerosols, because they very often contain BC, although in somewhat different fractions. The association of high levels of modeled BC with CALIPSO aerosols in general (see Zamora et al. (2017)) indicates that modeled BC is a fairly good proxy for strong (CALIPSO-detectable) aerosol layers during polar night, even though some local sources of combustion aerosols (Creamean et al., 2018; Maahn et al., 2017) might not be included in the model. Model comparisons to CALIPSO aerosol data in the study region also indicate that model-identified clean conditions ($BC < 30$ ng m$^{-3}$) are associated with significantly lower levels of CALIPSO aerosol layer presence relative to average or polluted conditions (see supplement for further details).

## 2.2 Cloud remote sensing observations

Cloud fraction is operationally defined based on the CloudSat algorithm, in CloudSat products available at a vertical resolution of 250 m. Cloud base and top heights were originally obtained from the 2B-GEOPROF-lidar product (Mace et al., 2009; Mace and Zhang, 2014), and the resulting cloud geometric thickness information was used to obtain a profile of vertical cloud fraction at FLEXPART-vertical resolutions for each observation point. All vertical cloud fraction profiles (numbering 10,422,219 total profiles over the Arctic Ocean) that fell within 12.5 km$^2$ stereographic projection grid cells (Cavalieri et al., 2014) were then averaged together. The gridded observations over sea ice (open ocean) include 15,999 (31,978) grid cells from fall, 43,687 (24,008) grid cells from winter, and 38,793 (15,289) grid cells from spring, with the observation numbers being a function of sea ice extent and length of darkness periods during each season.

Above 1 km, the 2B-GEOPROF-LIDAR product is similar to or better than ground-based observations, but cloud fraction can be underestimated by up to ~20% below ~1 km (Liu et al., 2017), indicating that cloud detection uncertainties in this study's lowest vertical bin (0.6-1.5 km) are higher there than in other altitude ranges.

CloudSat and CALIPSO do not sample north of 82°. The lack of data within this "pole hole" might mean that those sea ice regions are not well represented in this study. It is unclear how well the data outside the pole hole approximate the data inside it, as this region is the coldest part of the Arctic, and probably also contains some of the cleanest parts with respect to aerosols. Also note that thin-ice-cloud identification is particularly prone to errors over the Arctic, due in part to the widespread occurrence of sub-visible diamond dust and blowing snow. Additionally, the CloudSat radar can sometimes mistake

precipitation for clouds (de Boer et al., 2009), which can be particularly problematic under optically thick clouds that completely attenuate the CALIPSO lidar signal, and prevent lidar data from being collected below-cloud.

Cloud precipitation presence and phase were obtained from the CloudSat 2B-CLDCLASS-LIDAR version R04 (Wang, 2013).
This product captures precipitation with high confidence (Hudak et al., 2008). Phase determination has also been validated favorably at high latitudes (Barker et al., 2008), except that in some cases the radar can misclassify small liquid droplets as ice particles (Noh et al., 2011; Van Tricht et al., 2016). CloudSat may also fail to observe some ice and mixed-phase clouds below 1 km (Liu et al., 2017), suggesting higher uncertainties in cloud phase as well in the lowest vertical bin of this study. Here, cloud phase certainty values were required to be > 5, indicating a higher confidence in phase classification. If a FLEXPART-
resolution vertical bin contained clouds of different phases, that bin was excluded from phase-related portions of the analysis. As with CF data, nearby cloud precipitation and phase profile data were averaged within 12.5 km$^2$ grids at each altitude level prior to analysis.

Estimates of the longwave cloud radiative effect at the bottom-of-the-atmosphere ($CRE_{BOA}$) were obtained from the CloudSat
2B-FLXHR-LIDAR product, version R04 (Henderson et al., 2012; L'Ecuyer et al., 2008). This product is based on combined CloudSat, CALIPSO, and MODIS observations and time-coincident European Centre for Medium-Range Weather Forecasts (ECMWF) of atmospheric humidity, temperature and sea surface temperature, fed into the BugsRad two-stream, plane-parallel, doubling-adding radiative transfer model, following Henderson et al. (2012). Previous work shows that this product can severely underestimate downwelling LW radiation due to misclassification of small supercooled water as ice particles
(Van Tricht et al., 2016) leading to uncertainties in the absolute values of $CRE_{BOA}$. Here, we primarily focus on relative differences in $CRE_{BOA}$ between two subsets of data: those with high and low modeled BC values. The uncertainty due to misclassification of small particle phase is similar in both subsets of data, which are collected over the same surfaces and years, allowing for meaningful comparisons to be made between the two datasets despite uncertainty in the absolute values.

**2.3 AIRS observations**

Air mass temperature and relative humidity at pressure levels ranging from 1000-250 hPa were obtained from the Atmospheric Infrared Sounder (AIRS) level 3 version 6 dataset (Susskind et al., 2014) on the descending orbit (collected at 1:30 am local time). The AIRS instrument provides quality controlled, accurate daily observations over the full study area, including during nighttime conditions, and is validated for use over the Arctic (Boisvert et al., 2015). Data are available in most cloud conditions, although data are not available in completely cloud-covered conditions. Level 3 data, which average observations over a 1x1º
horizontal grid and report at 20 vertical pressure levels, are used instead of level 2 data to obtain closest approximate T and RH data when cloud fractions are high. Errors in this product are highest at larger cloud fractions and below optically thick clouds. For comparison to other datasets in this study, AIRS data were averaged into the coarser FLEXPART model vertical resolution.

## 2.4 Data analysis

Differences in relative humidity, temperature, and 12.5 km$^2$ gridded CF ($\overline{dRH}$, $\overline{dT}$, and $\overline{dCF}$, respectively) between all ($\overline{RH}$, $\overline{T}$, and $\overline{CF}$, respectively) and clean ($\overline{RH_c}$, $\overline{T_c}$, and $\overline{CF_c}$, respectively) conditions were calculated over sea ice and open ocean regions as follows:

(1) $\overline{dRH} = \overline{RH} - \overline{RH_c}$

(2) $\overline{dT} = \overline{T} - \overline{T_c}$

(3) $\overline{dCF} = \overline{CF} - \overline{CF_c}$

In a process conceptually fairly similar to previous work (Chen et al., 2014) (see Figure 1 as an example), spatially gridded CF and BC data in all and clean (BC < 30 ng m$^{-3}$) conditions were sorted into 5% relative humidity bins and 5 °C temperature bins, and then the differences in all conditions minus clean conditions were compared within each T-RH bin (dCF$_{T,RH}$ and dBC$_{T,RH}$, respectively). We then compared the differences in all minus clean conditions within each T-RH bin for the change in cloud fraction (dCF$_{T,RH}$) black carbon (dBC$_{T,RH}$), cloud phase (dCP$_{T,RH}$) and precipitation frequency (dpptn$_{T,RH}$). Data were analyzed separately over sea ice vs. open ocean, and within different altitude layers.

Stratifying by similar T and RH conditions isolates some of the systematic BC co-variability with T and RH, helping clarify the BC aerosol impact on cloud fraction. The 5 °C T and 5% RH bin increments were chosen to balance the benefits of larger sample sizes against the drawbacks of reduced bin representativeness at smaller bin sizes. Data plotted at larger bin increments resulted in similar trends (data not shown).

The estimated microphysical impact of combustion aerosols on total CF over the Arctic Ocean during polar night is calculated from the mean dCF$_{T,RH}$, weighted by the number of 12.5 km$^2$ grids containing observations falling within each RH and T bin, abbreviated as $\overline{dCF_{T,RH}}$. Averaging over sea ice and open ocean regions helps reduce the effects of horizontal winds on factors such as aerosol advection, which can co-vary on local scales with aerosols (Engström and Ekman, 2010; Nishant and Sherwood, 2017). That, in combination with accounting for variations in the T and RH data, enables us to capture several key meteorological parameters that might influence cloud fraction. However, there are no reliable space-borne measurements for vertical velocity, which might also co-vary systematically with BC on a regional scale, and meteorological reanalyses of large-scale vertical motion over the wintertime Arctic are not well-validated (Jakobson et al., 2012). Our focus on nighttime data over the flat ocean surface likely reduces effects from solar-heating-driven vertical motion, but the full uncertainty from this parameter cannot be accounted for here. For example, if cold aerosol-containing continental air is routinely advected over

warmer open ocean areas, that could induce systemic convection (Serreze and Barry, 2005) that might not be fully captured by the T and RH stratification. To provide at least some generalized grouping of clouds likely to be influenced by different large-scale vertical motion, we analyzed altitude layers and surface types (sea ice and open ocean) separately.

## 3 Results

### 3.1 Aerosol microphysical effects on cloud fraction

Systematic regional co-variability of aerosols and meteorological factors must be accounted for in order to avoid overestimating aerosol impacts on clouds (Coopman et al., 2018; Gryspeerdt et al., 2016). To illustrate this point, Figure 2 shows the longwave $CRE_{BOA}$ for the upper and lower quartiles of FLEXPART model column BC concentrations, calculated during the entire study period. The upper and lower quartile ranges of column BC levels are associated with very large (~10

W m$^{-2}$) differences in median longwave $CRE_{BOA}$ over sea ice (Fig. 2). This value is estimated from the median difference in 12.5 km$^2$ gridded $CRE_{BOA}$ values over sea ice regions across the Arctic Ocean during the study period, in grid cells with a minimum of at least three observations in the upper and lower quartile ranges of column BC levels. However, when we compare the median relative humidity and temperature profiles with column BC levels in the upper quartile over sea ice (Fig. 2f, red lines) and open ocean (Fig. 2g, red lines) to the lower quartile profiles (blue lines, same figures), it is clear that column

BC levels over sea ice are also associated with noticeable differences in median relative humidity and temperature profiles (Fig. 2f). Small differences in lower tropospheric stability (Fig. 2e), defined as the difference in potential temperature between 700 and 1000 hPa, are also observed. These meteorological factors strongly affect CF and CP, which in turn help drive $CRE_{BOA}$. As a result, aerosol microphysical effects may contribute to only a fraction of the $CRE_{BOA}$ differences shown in Figure 2.

To help better understand co-varying meteorological effects on CF specifically, we assessed a generalized additive model (GAM) (Hastie and Tibshirani, 1990) of the $\overline{dRH}$, $\overline{dT}$, and $\overline{dCF}$ data at each vertical level, season, and surface type (Table 1). Seasonal differences in solar illumination, sea ice extent, and BC levels led to some sample number differences for sea ice and open ocean at different times of the year (Table 2). In the GAM, the seasonal values in Table 1 were weighted equally to represent the equal periods of the year being sampled.

The GAM suggests that co-varying differences in $\overline{dRH}$ and $\overline{dT}$ by themselves can explain up to 91% of the variability in $\overline{dCF}$ (as measured by deviance, a statistic similar to variance (Jorgensen, 1997)). Because aerosols can co-vary with T and RH (e.g., because polluted air masses are more likely to have recently resided near the continental surface than clean air masses), aerosols could be responsible for some of this explained variability even without being explicitly included in this GAM. For instance,

a GAM based only on $\overline{dBC}$ explains up to 40% of $\overline{dCF}$ variability. A GAM containing $\overline{dBC}$, $\overline{dRH}$, and $\overline{dT}$ explains 97% of the $\overline{dCF}$ variability, and thus a lower range on the influence of temperature and relative humidity on differences in CF from would be ~57% (97% minus 40%). The finding that co-varying temperature and relative humidity explain most (57-91%) of the $\overline{dCF}$

variability underscores the importance of interpreting aerosol effects on clouds in the context of co-varying temperature and relative humidity. It also indicates that changes in T and RH of air masses entering the Arctic could have important impacts on observed CF, to a degree that is likely to be much more regionally important than the microphysical effects of the aerosols themselves.

Cloud fraction substantially differed among all and clean conditions for many combinations of T, RH, altitude and surface type (Fig. 1). Estimated aerosol impacts on total CF depend on altitude and surface type, but are fairly consistent among seasons (Figs. S1-S3). At the lowest levels (0.6-2.5 km over sea ice and 0.6-1.5 km over open ocean), weighted mean $dCF_{T,RH}$ ($\overline{dCF_{T,RH}}$) is negative, resulting in an ~6% reduction in CF relative to clean conditions over sea ice (-0.6% over open ocean)

(Fig. 3). In contrast, $\overline{dCF_{T,RH}}$ between 4-8.5 km elevation increased by 3-5% over both surfaces, indicating more cloud cover at high altitudes for combustion aerosol influenced clouds compared to clean conditions. Absolute $\overline{dCF_{T,RH}}$ changes over sea ice and open ocean ranged between -1.7% to 0.7% and -0.2 to 1.4%, respectively, depending on altitude. Note that the $\overline{dCF_{T,RH}}$ value is based on all $dCF_{T,RH}$ data, including those from T and RH ranges where $dCF_{T,RH}$ is not significantly different from zero (i.e., as indicated by the white Xs in Figure 1). Including all data avoids biasing the results in favor of the meteorological

conditions where $dCF_{T,RH}$ is most observable.

The $dCF_{T,RH}$ and $dBC_{T,RH}$ relationships (Fig. 4) indicate that there was more cloud cover in slightly polluted conditions but less cloud cover at higher $dBC_{T,RH}$ levels (> 10-20 ng m$^{-3}$) relative to clean conditions. $dCF_{T,RH}$ declined significantly at $dBC_{T,RH}$ > 20 ng m$^{-3}$ within most single altitude layers over sea ice, and over open ocean at 1.5-2.5 km (Fig. 3).

Note that the fall period typically has cleaner and warmer conditions compared to winter and spring (Table 1), which tend to occur more heavily on opposite sides of the scatter plots for each altitude layer in Figure 4. Thus, any large, systematic differences in the vertical winds between fall and spring could influence the outermost points within individual altitude layers, and it is not easy to control for this effect. However, the trends among altitude layers show that $dCF_{T,RH}$ is essentially identical

over sea ice and open ocean at low $dBC_{T,RH}$ values, which occur mostly at high altitudes. Also, $dCF_{T,RH}$ changes at high $dBC_{T,RH}$/low altitude are more observable over sea ice (Fig. 4), where lower tropospheric stability was greater and temperatures were colder (Figs. 2e-g). Previous studies have also observed more apparent aerosol microphysical effects under more stable conditions in the Arctic (Coopman et al., 2018; Zamora et al., 2017). Possible reasons for the disparate behavior at different altitudes are discussed below.

**3.2 Aerosol microphysical effects on cloud phase partitioning**

Weighted mean differences in CP partitioning between all minus clean conditions within the same T and RH bins ($\overline{dCP_{T,RH}}$) are discussed for clouds between 0.6-4 km, because clouds at higher altitudes occur mostly in the ice phase (Fig. 5a)

(Devasthale et al., 2011a; Liu et al., 2017). Over sea ice between 0.6-4 km, all air masses contained a higher relative fraction of ice phase clouds (IPCs) and a lower relative fraction of liquid phase clouds (LPCs) and mixed phase clouds (MPCs) relative to clean air masses (Fig. 3). This effect was significant up to 4 km (paired Wilcoxon rank test (Sokal and Rohlf, 1995), $p <$ 0.05), except in LPCs between 2.5-4 km, where lower sample numbers might obscure any changes. Changes in phase partitioning over the sea ice region varied between -4.2 and 6.5%, depending on altitude and phase (Fig. 3). From Figure 3, over sea ice between 1.5-2.5 km, the relative contributions of LPCs and MPCs were significantly lower at high $dBC_{T,RH}$ levels ($>20$ ng m$^{-3}$), whereas that of IPCs was significantly higher. No significant relationships with $dBC_{T,RH}$ were observed above or below that altitude, although higher BC and CP uncertainties near the surface might mask weak signals in that altitude range. The reduction in liquid-containing clouds at higher $dBC_{T,RH}$ levels over sea ice is consistent with a "glaciation effect" (Lohmann, 2002), whereby increased presence of aerosols leads to ice particle formation and cloud dissipation, as observed in section 3.1.

Over open ocean, significant changes in $\overline{dCP_{T,RH}}$ were observed less frequently (Fig. 3), and they tended to be smaller than over sea ice (absolute values < 2%). The relative fraction of liquid clouds was reduced between 0.6-1.5 km (Fig. 3), where LPC fractions were highest (Fig. 5a). However, unlike over sea ice, the relative fraction of MPCs over open ocean increased (though not significantly so between 1.5-2.5 km), and that of IPCs decreased (significant only between 2.5-4 km). The reason for the different effects on ice-containing clouds over sea ice and open ocean is unclear, although the higher temperatures may play a role.

### 3.3 Aerosol microphysical influences on precipitation frequency

Differences in precipitation frequency, $dpptn_{T,RH}$, reflect aerosol microphysical impacts on: 1) the frequency of precipitation within a specific air volume, and 2) the relative likelihood of individual cloud phases within that air volume to be precipitating. We analyse the difference in precipitation frequency; however, an analysis of total precipitation amounts or precipitating particle microphysics is beyond the scope of this study.

Based on weighted mean $dpptn_{T,RH}$ values ($\overline{dpptn_{T,RH}}$), estimated aerosol microphysical effects on regional precipitation frequency were small but significant at many altitudes (Fig. 3, Fig. S4). In all air mass conditions, precipitation frequency was 1.2-3.1% higher below 6 km over open ocean and below 1.5 km over sea ice relative to clean conditions, depending on altitude (Fig. 3). In contrast, clean clouds between 6-8.5 km over open ocean were slightly more likely to be precipitating ($\overline{dpptn_{T,RH}}$ ~ -1%).

Over sea ice, ~94% of MPCs were present below 4 km (Fig. 5a). In these MPCs, $\overline{dpptn_{T,RH}}$ was positive (~ +1%), indicating slightly more frequent precipitation on average in all vs. clean MPCs (Fig. 5b). Significant differences between all and clean

conditions were not observed for $\overline{\text{dpptn}_{T,RH}}$ in IPCs or LPCs over sea ice, except for a slight (-0.4%) reduction in $\overline{\text{dpptn}_{T,RH}}$ in ice clouds at 6-8.5 km (Fig. 5b). However, significant rank correlations (Kendall's tau coefficient = 0.3, p < 0.05) indicate that higher $dBC_{T,RH}$ values were associated with slightly more frequent IPC precipitation over sea ice between 0.6-1.5 km (also see Fig. S5). We observed no strong link between $dBC_{T,RH}$ and $dpptn_{T,RH}$ at other locations/altitudes.

As over sea ice, MPC $\overline{\text{dpptn}_{T,RH}}$ was slightly positive (≤1%) below 4 km over open ocean (Fig. 5b), indicating slightly more MPC precipitation in all vs. clean conditions. IPC $\overline{\text{dpptn}_{T,RH}}$ was also slightly positive between 1.5-4 km, whereas liquid cloud $\overline{\text{dpptn}_{T,RH}}$ was slightly negative between 1.5-4 km. Between 6-8.5 km over open ocean, the $\overline{\text{dpptn}_{T,RH}}$ in MPCs was slightly negative at ∼-1% (Fig. 5b).

Based on single or small cloud samples, others have observed decreased precipitation probability with increased aerosol concentrations in Arctic MPCs (Lance et al., 2011; Morrison et al., 2008). It is not entirely clear why the large-scale, regional trends observed here appear to be opposite these smaller-scale *in situ* observations, but recent work indicates that aerosols might influence ice content of the clouds, and thereby affect precipitation (Fu and Xue, 2017; Norgren et al., 2018; Zamora et al., 2017) and potentially CF. Perhaps lower total CF below 4 km leads to less frequent precipitation in these air volumes over the Arctic. The higher MPC precipitation probability and lower MPC cover (as indicated by reduced MPC relative fraction of total CF) at higher aerosol concentrations support this hypothesis.

## 4 Discussion – Potential aerosol microphysical mechanisms

Specific aerosol-cloud microphysical mechanisms are difficult to identify with confidence from space-borne measurements alone, but some possibilities can be explored. At high altitudes (6-8.5 km) over sea ice, $\overline{\text{dCF}_{T,RH}}$ was higher and $\overline{\text{dpptn}_{T,RH}}$ was lower in all vs. clean conditions (Fig. 3), supporting the hypothesis that aerosols are modifying cloud properties on a regional (i.e., sea ice and open ocean) scale at these altitudes, even though the net changes were relatively small. These modifications to predominantly IPCs at high altitude likely involve aerosol effects on ice crystal formation or properties. Oreopoulos et al. (2017) similarly reported global-scale increases in ice CF with higher aerosol concentration in their CR1 cases (typically high ice clouds of small optical thickness over the tropics), which was linked to reduced ice cloud effective radius.

There are several mechanisms by which aerosols might modify ice crystal number or size that could cause the observed changes in precipitation and CF in the 6-8.5 km range. Although BC itself is not a good source of INPs (Vergara-Temprado et al., 2018), combustion aerosols associated with BC might act as ice nucleating particles (INPs) (Kanji et al., 2017) at the extreme cold temperatures found at high-altitude Arctic polar night. This could potentially lead to smaller, more numerous ice particles that precipitate less (Lohmann and Feichter, 2005), in line with our observations, although some models suggest that INPs may

instead lead to larger ice crystals in cirrus clouds compared to homogeneous freezing (Heymsfield et al., 2016). Alternatively, combustion aerosols might reduce the efficiency of pre-existing INPs through the "deactivation effect" (Archuleta et al., 2005; Cziczo et al., 2009). Reduced ice crystal formation rates could then lead to more frequent air mass saturation with respect to liquid water, more water droplets that freeze homogeneously, and smaller, more numerous ice particles, and less precipitation

5 (Girard et al., 2013) as observed here. This effect could lead to enhanced total CF over the Arctic (Du et al., 2011). Although absolute humidity within the different T-RH bins between 6-8.5 km is not systematically related to higher $dBC_{T,RH}$ levels as one might expect with the deactivation effect, it is possible that pre-sorting the data by 5% RH bins to reduce the impacts of meteorological co-variation could make evidence for this effect more difficult to observe. Therefore, it is difficult to say whether this study is consistent with the deactivation effect hypothesis, but it does not preclude it.

The specific microphysical mechanisms affecting lower altitude clouds are more challenging to identify without in situ data due to the high prevalence of liquid-containing clouds (Fig. 5a). Combustion aerosols can affect precipitation rates by changing droplet numbers and sizes, and thereby possibly collision and coalescence (Albrecht, 1989), riming (Lohmann and Feichter, 2005; Saleeby et al., 2009), or freezing (Bigg, 1953). If these aerosols affect INP levels, they could also affect ice nucleation

rates and ice particle concentrations, leading to MPC and LPC glaciation, enhanced precipitation, and reduced cloud cover (the "glaciation effect"). Our observations do support the possibility of a glaciation effect, because once T and RH co-variability are accounted for, all air masses at low altitudes (0.6 to 1.5-2.5 km) have lower CF compared to clean conditions. They also have more frequent precipitation in IPCs at high $dBC_{T,RH}$, and a higher relative fraction of IPCs over sea ice and MPCs over the warmer open ocean. Each of these changes is significantly different between low and high $dBC_{T,RH}$

concentrations at a variety of altitudes and surface types (Fig. 3), suggesting that aerosols may help convert liquid droplets to larger ice particles that precipitate out and reduce CF in lower altitude clouds.

These observations are in-line with other studies indicating that aerosols can dissipate Arctic MPCs (Fu and Xue, 2017; Norgren et al., 2018) and increase their precipitation (Kravitz et al., 2014; Morrison et al., 2011). Assuming they act as INPs,

various modelling studies and a remote sensing study also suggest that aerosols can reduce liquid water path or supercooled water frequency (Fan et al., 2009; Morrison et al., 2011; Ovchinnikov et al., 2014; Pinto, 1998; Tan et al., 2014). The observations over sea ice contrast with some model predictions that MPCs should increase in more polluted conditions through the deactivation effect (Du et al., 2011; Girard et al., 2005, 2013). They also contrast with a previous remote sensing study (Zamora et al., 2017) indicating that thin and predominantly liquid Arctic Ocean clouds are more likely to be the liquid phase

at high BC concentrations. However, the clouds in that study may not be fully comparable, as they constitute only ~5% of the cloud types in this study. Note that shortwave processes might alter how aerosols impact mixed phase CF during daytime (Solomon et al., 2015), and any such effects would not be observed in the current, nighttime study. Changes in higher altitude clouds might also change underlying cloud properties through a seeding effect, which could impact cloud properties at lower altitudes.

# 5 Summary and conclusions

Upper quartile levels of total column BC (a proxy for combustion aerosols) are associated with very large (~10 W m$^{-2}$) differences in longwave cloud radiative effects at the sea ice surface compared to the lower quartile column BC levels. However, relative humidity in particular over sea ice is very different in the two aerosol conditions, which likely drives much of the CRE$_{BOA}$ differences in Figure 2. The CRE$_{BOA}$ is impacted to a high degree by CF. We found that BC predicted at most 40% of the observed differences in sea ice and open ocean CF between all and clean conditions in the altitude ranges of interest in this study (0.6-8.5 km), whereas AIRS-derived temperature and relative humidity predicted 57-91% of these differences. These observations indicate that changes in T and RH of air masses entering the Arctic will likely have a more regionally important influence on observed CF than the microphysical effects of the aerosols themselves, although aerosols are not to be discounted. In line with previous studies (e.g., Gryspeerdt et al. (2016); Coopman et al. (2018)), these results also underscore the need for large sample volumes to identify systematic air mass differences between clean and all conditions, and a way to reduce the confounding effects of meteorological co-variation on these samples. To accomplish this, we analyzed over 10 million profiles across the Arctic Ocean, which were binned into similar T and RH groups. We analyzed the data separately over sea ice vs. open ocean, and within different altitude layers.

In general, combustion aerosol microphysical effects were most observable where the highest aerosol effect would be expected: a) at lower altitudes where aerosol concentrations are often higher (Devasthale et al., 2011b) and b) over sea ice, where atmospheric stability is greater, and aerosol microphysical effects on clouds are less likely to be overwhelmed by meteorological factors such as high vertical velocity. Relative to clean conditions, low clouds over sea ice had ~6% smaller CF and 3% higher precipitation frequency, whereas at high altitudes, CF increased by 4% and precipitation was 2% less frequent. Similar but smaller trends in CF and low-altitude precipitation were observed over open ocean. Below 1.5 km, we also observed a 7% reduction in the LPC and MPC fractions over sea ice, but a slight increase in MPCs relative to LPCs over open ocean. The different effect on MPCs over sea ice and open ocean may be related to the higher temperatures over open ocean, leading to less efficient ice formation, or to some other, as yet unknown, factor. Observations from others (e.g., Chernokulsky et al. (2017); Eastman and Warren (2010)) show that expansion of open ocean areas appear to be connected to changing Arctic Ocean cloud properties. The different cloud responses to aerosols that we observe over sea ice vs. open ocean may provide partial clues into the cause of this behaviour, and into the future impacts of combustion aerosols on the Arctic system in general.

These results are subject to various uncertainties, including possible confounding influences from large-scale vertical motion that is difficult to measure in situ, any systematic model errors in identifying aerosol layers at the right altitude (see supplementary information), and the uncertain relationships between modeled BC and INP and cloud condensation nuclei (CCN) concentrations. Associations between dBC$_{T,RH}$ and cloud properties may be difficult to observe given the low BC

concentrations at altitudes > 4 km, and may be further masked by complex aerosol-cloud relationships. Despite these uncertainties, significant differences in $dCF_{T,RH}$, $dpptn_{T,RH}$, and $dCP_{T,RH}$ at high $dBC_{T,RH}$ concentrations provide evidence that aerosol microphysical effects were driving the observed patterns, as opposed to some other factor. Furthermore, these observations leave open the possibility that other cloud property relationships with $dBC_{T,RH}$ exist, but are not observable with

the available data.

The mechanisms responsible for these changes cannot be fully elucidated from modeling and remote sensing data alone. The observed increases in CF and decreases in precipitation at 6-8.5 km likely involve aerosol effects on ice crystal formation or properties, given that nearly all of these clouds are in the ice phase. These effects might include a deactivation of pre-existing

INPs, or conversely an enhancement of INPs by combustion aerosols themselves at the very low temperatures observed at these high altitudes during Arctic polar night. The reduced CF, more frequent precipitation in mixed phase clouds, and reduced relative fraction of mixed (liquid) phase clouds over sea ice (open ocean) seem to point towards aerosols either participating in the conversion of liquid droplets to larger ice particles that precipitate and reduce CF, similar to a glaciation effect, or potentially to their impacts on precipitation in higher clouds, changing underlying cloud properties through a seeding effect.

Further focused studies on these mechanisms would be of great interest, along with targeted aircraft measurements of the relevant aerosol and cloud properties, providing greater detail at higher spatial and temporal resolution. To improve quantification of Arctic aerosol-cloud microphysical interactions from space, two major uncertainties also require better quantification: (1) large-scale vertical motion and (2) altitude-resolved aerosol amount and type information. Obtaining more ground-based observations of clouds lower than 0.6 km, which are radiatively significant but not measured well by satellite,

is also important.

**Data availability.** CloudSat data were obtained from http://www.cloudsat. cira.colostate.edu/order-data/, and AIRS data were obtained from https://disc.gsfc.nasa.gov/. For access to the CALIPSO, ETOPO and NSIDC data, see CALIPSO Science Team (2016), Amante and Eakins (2009), and Meier et al. (2013), respectively. The data underlying Figures 1, 2,

and 4 are presented in text files in the supplementary material, and the data from Figures 3 and 5 are presented tabularly in Table S1.

**Author contributions.** LMZ., RAK, and KBH designed the study and the statistical analysis. LMZ processed the satellite data, and AS and SE processed FLEXPART model output. LMZ wrote the manuscript. All authors contributed to revising the manuscript.

**Competing interests.** The authors declare that they have no conflict of interest.

**Acknowledgements**

The authors would like to thank A. Ackerman, G. De Boer, J. Creamean, A. Fridlind, B. Hegyi, K.-M. Kim, T. L'Ecuyer, J. Limbacher, G. McFarquhar, L. Oreopoulos, and J. Susskind for helpful discussions, as well as two anonymous reviewers.

CALIPSO data were provided by the NASA Langley Research Center Atmospheric Science Data Center, and the CIRA CloudSat Data Processing Center provided the CloudSat data. The work of LMZ and RAK is supported in part by NASA's Climate and Radiation Research and Analysis Program under H. Maring and NASA's Atmospheric Composition Program under R. Eckman and K. Jucks.

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

**Tables**

**Table 1.** The mean temperature (T, °C), relative humidity (RH, %), black carbon (BC, ng m$^{-3}$), and cloud fraction (CF, %) observed over the Arctic Ocean study region over sea ice and open ocean during different seasons and altitude levels. Data are shown for all conditions. Also shown are the mean differences between all minus clean conditions ("difference") for T, RH, BC, and CF, referred to as $\overline{dT}$, $\overline{dRH}$, $\overline{dBC}$, and $\overline{dCF}$ in the text.

| | Altitude | Sea ice | | | | | | Open ocean | | | | | |
| | | Fall (ASO) | | Winter (NDJ) | | Spring (FMA) | | Fall (ASO) | | Winter (NDJ) | | Spring (FMA) | |
| | levels (km) | all | difference | all | difference | all | difference | all | difference | all | difference | all | difference |
|---|---|---|---|---|---|---|---|---|---|---|---|---|---|
| Temperature (°C) | 6-8.5 | -48.7 | 0.0 | -53.0 | 0.0 | -51.9 | 0.1 | -45.0 | 0.1 | -49.6 | 0.0 | -49.2 | 0.1 |
| | 4-6 | -32.3 | 0.5 | -38.1 | 0.2 | -40.0 | 0.4 | -27.7 | 0.3 | -33.2 | 0.1 | -35.5 | 0.1 |
| | 2.5-4 | -21.6 | 0.5 | -27.5 | 0.1 | -29.5 | 0.1 | -15.8 | 0.4 | -21.1 | 0.2 | -23.6 | 0.0 |
| | 1.5-2.5 | -15.5 | 0.5 | -21.6 | -0.4 | -23.4 | -0.7 | -9.2 | 0.4 | -14.6 | 0.4 | -16.6 | -0.2 |
| | 0.6-1.5 | -11.3 | 0.4 | -17.7 | -0.6 | -19.6 | -1.3 | -4.5 | 0.3 | -9.9 | 0.4 | -11.6 | -0.3 |
| | | | | | | | | | | | | | |
| Relative Humidity (%) | 6-8.5 | 74.6 | 0.0 | 65.6 | 0.4 | 50.4 | 1.0 | 69.0 | 0.1 | 63.8 | 0.2 | 50.9 | 0.4 |
| | 4-6 | 61.5 | -0.3 | 62.7 | 0.4 | 59.8 | -0.6 | 57.1 | -0.1 | 57.6 | 0.6 | 58.5 | 0.9 |
| | 2.5-4 | 66.7 | -0.6 | 69.3 | 0.2 | 66.2 | -1.8 | 61.4 | 0.2 | 58.6 | 1.6 | 57.1 | 0.6 |
| | 1.5-2.5 | 76.9 | -0.9 | 74.0 | -0.4 | 68.2 | -0.3 | 74.5 | 0.2 | 72.5 | 1.9 | 69.1 | 0.5 |
| | 0.6-1.5 | 85.1 | -0.3 | 78.1 | -2.2 | 70.6 | 0.2 | 87.2 | -0.3 | 88.6 | 1.7 | 84.4 | -0.2 |
| | | | | | | | | | | | | | |
| BC (ng m$^{-3}$) | 6-8.5 | 14 | 1 | 13 | 2 | 19 | 5 | 14 | 2 | 11 | 1 | 17 | 3 |
| | 4-6 | 17 | 3 | 22 | 6 | 30 | 11 | 17 | 3 | 18 | 3 | 28 | 9 |
| | 2.5-4 | 19.8 | 4 | 31 | 13 | 41 | 19 | 19 | 5 | 28 | 10 | 38 | 16 |
| | 1.5-2.5 | 20 | 4 | 39 | 21 | 53 | 27 | 22 | 7 | 43 | 23 | 47 | 22 |
| | 0.6-1.5 | 17 | 3 | 51 | 31 | 66 | 32 | 22 | 9 | 58 | 34 | 56 | 27 |
| | | | | | | | | | | | | | |
| Mean CF (%) | 6-8.5 | 15.4 | 0.2 | 15.0 | 0.4 | 9.1 | 0.9 | 19.3 | 1.0 | 22.3 | 0.4 | 15.3 | 0.5 |
| | 4-6 | 22.7 | 0.6 | 21.8 | 1.3 | 16.8 | 0.1 | 24.3 | 0.8 | 26.0 | 0.8 | 22.7 | 0.7 |
| | 2.5-4 | 29.1 | 1.0 | 25.9 | 1.0 | 20.8 | -2.1 | 27.9 | 1.0 | 28.8 | 2.2 | 27.4 | 0.5 |
| | 1.5-2.5 | 35.0 | -0.3 | 30.3 | -0.4 | 24.0 | -3.5 | 34.3 | 1.4 | 39.4 | 2.1 | 39.5 | 0.3 |
| | 0.6-1.5 | 33.3 | -0.2 | 26.9 | -2.0 | 22.0 | -3.0 | 35.0 | -0.3 | 43.0 | -0.4 | 43.6 | -0.3 |

**Table 2.** Total profile numbers during each season of the study over sea ice and open ocean regions. Also shown are the percent of samples determined to be clean (BC < 30 ng m$^{-3}$) at different altitudes. Seasonal differences in sample numbers depend on factors such as solar illumination, sea ice extent, and, for clean samples, seasonal variations in BC levels.

| | | Sea ice | | | Open ocean | | |
|---|---|---|---|---|---|---|---|
| | | Fall (ASO) | Winter (NDJ) | Spring (FMA) | Fall (ASO) | Winter (NDJ) | Spring (FMA) |
| Total samples | | 457,504 | 4,687,541 | 1,757,034 | 1,153,806 | 1,429,840 | 529,904 |
| | Altitude levels (km) | | | | | | |
| | 6-8.5 | 94% | 94% | 85% | 92% | 97% | 89% |
| % "Clean" | 4-6 | 88% | 81% | 68% | 88% | 88% | 69% |
| samples | 2.5-4 | 85% | 65% | 54% | 83% | 70% | 59% |
| | 1.5-2.5 | 83% | 51% | 39% | 81% | 54% | 50% |
| | 0.6-1.5 | 88% | 46% | 27% | 83% | 50% | 48% |

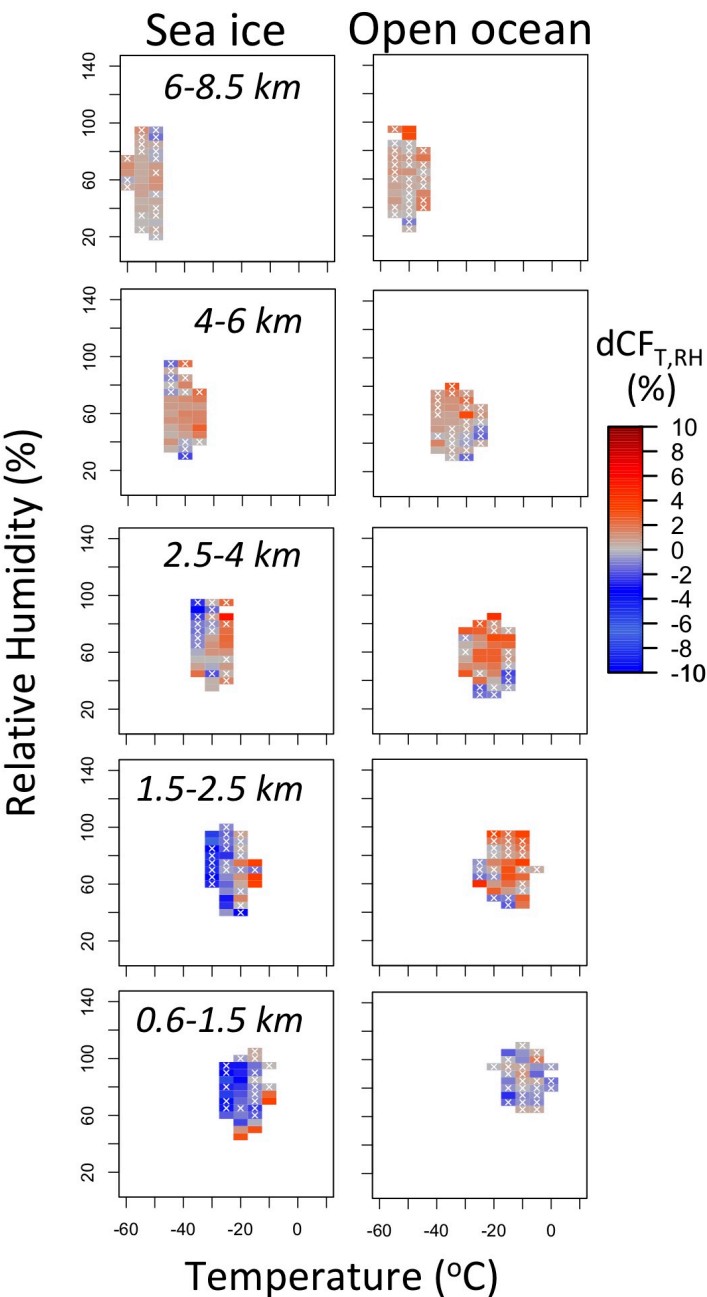

**Figure 1: An example of dCF$_{T,RH}$ output at each altitude level. For illustration purposes, here each grid cell represents ≥ 7500 km$^2$ of gridded observations. Blue and red colors indicate negative and positive dCF$_{T,RH}$, respectively. A white X indicates that the cell value is not significantly different from zero (Wilcoxon rank test, $p < 0.05$). Note that each underlying Wilcoxon rank test has a 5% chance of yielding a false positive indication of statistical significance or an unknown (but likely much higher) chance of yielding a false negative result. Consequently, the distribution of Xs should not be over-interpreted. The number of Xs, however, provides an objective way to test whether the evidence for an effect on the grid as a whole is significant. This is consistently the case; in all panels, individually significant cells numbered more than expected at random (binomial test, $p < 0.001$).**

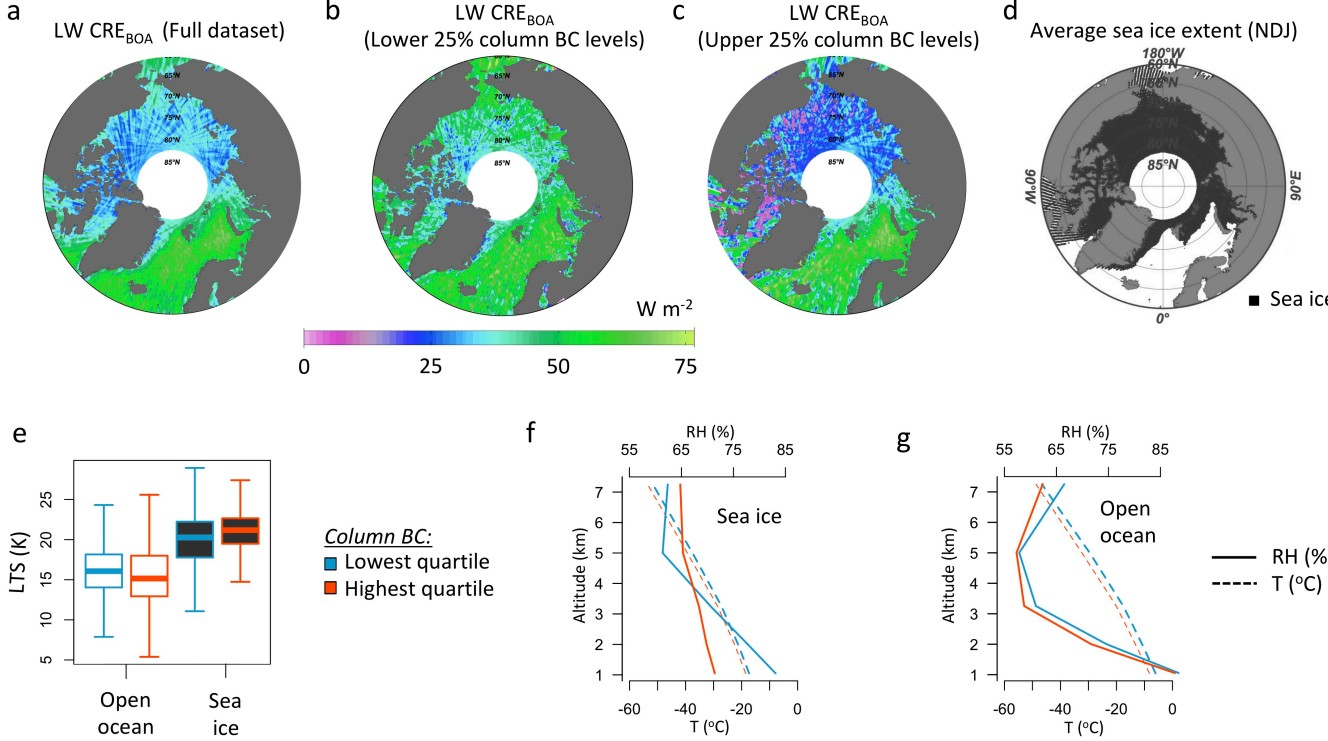

**Figure 2: a-c)** CloudSat FLXHR-LIDAR longwave cloud radiative effect at the bottom of atmosphere (i.e., surface) (LW CRE$_{BOA}$) during polar night for a) the full dataset, and the subsets of data containing the lower (b) and upper (c) quartiles of modeled column BC concentrations. For reference, d) shows the average winter (November to January) sea ice extent up to 82°N. Also shown are e) boxplots of the lower tropospheric stability (LTS, K), and f, g) the median temperature (T, °C) and relative humidity (RH, %) for the lower (blue) and upper (red) quartile column BC concentrations over open ocean and sea ice. All differences in e-g) are significant (p < 0.0001), based on a Wilcoxon rank test.

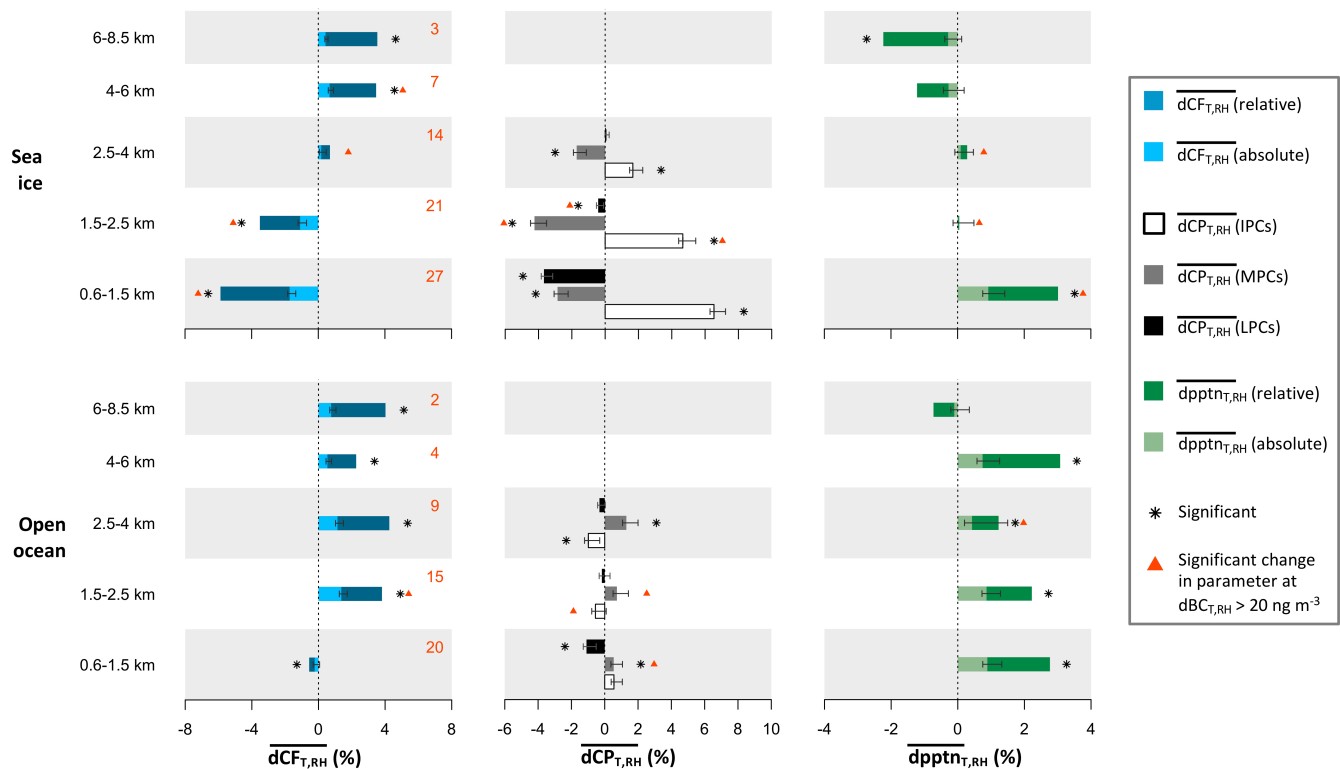

**Figure 3: Summary of $\overline{dCF_{T,RH}}$, $\overline{dCP_{T,RH}}$ (up to 4 km, %), and $\overline{dpptn_{T,RH}}$ (%) in different altitude ranges over sea ice and open ocean. For $\overline{dCF_{T,RH}}$ and $\overline{dpptn_{T,RH}}$, different color bars (overlaid, not stacked) show the absolute change within the air volume of interest (lighter colors) and the relative percent change with respect to the value found in clean conditions (darker colors). The $\overline{dBC_{T,RH}}$ values (ng m$^{-3}$) are presented for each altitude (red, upper right in the left two panels). An asterisk (\*) indicates that the differences between all and clean conditions were significant for both relative and absolute values, based on a paired Wilcoxon rank test, p < 0.05, using T-RH grid cells containing > 800 (400 for $\overline{dCP_{T,RH}}$) 12.5-km$^2$ gridded observations. Values marked by a red triangle indicate a significant change in the parameter where dBC$_{T,RH}$ > 20 ng m$^{-3}$ (Wilcoxon rank test, p < 0.05). Error bars show bootstrapped 95% confidence intervals for the weighted mean.**

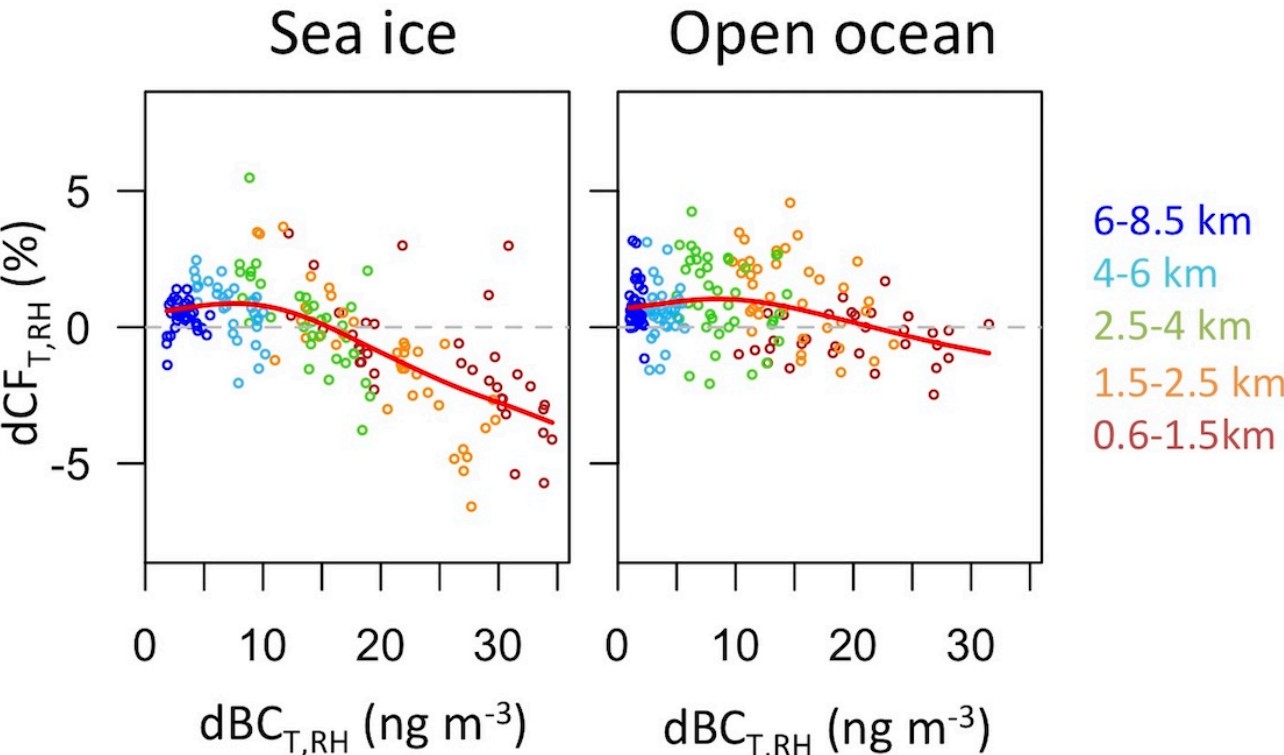

**Figure 4: The relationship between $dCF_{T,RH}$ (%) and $dBC_{T,RH}$ (ng m$^{-3}$) over sea ice and open ocean at different altitude levels (color coded) for the points plotted in Figure 1. The red line is a cubic smoothing spline of the data among all altitudes. In order to avoid obscuring emergent properties of the full dataset, the data include all meteorological conditions, including those where $dCF_{T,RH}$ are not significantly different from zero (as noted by white Xs in Figure 1).**

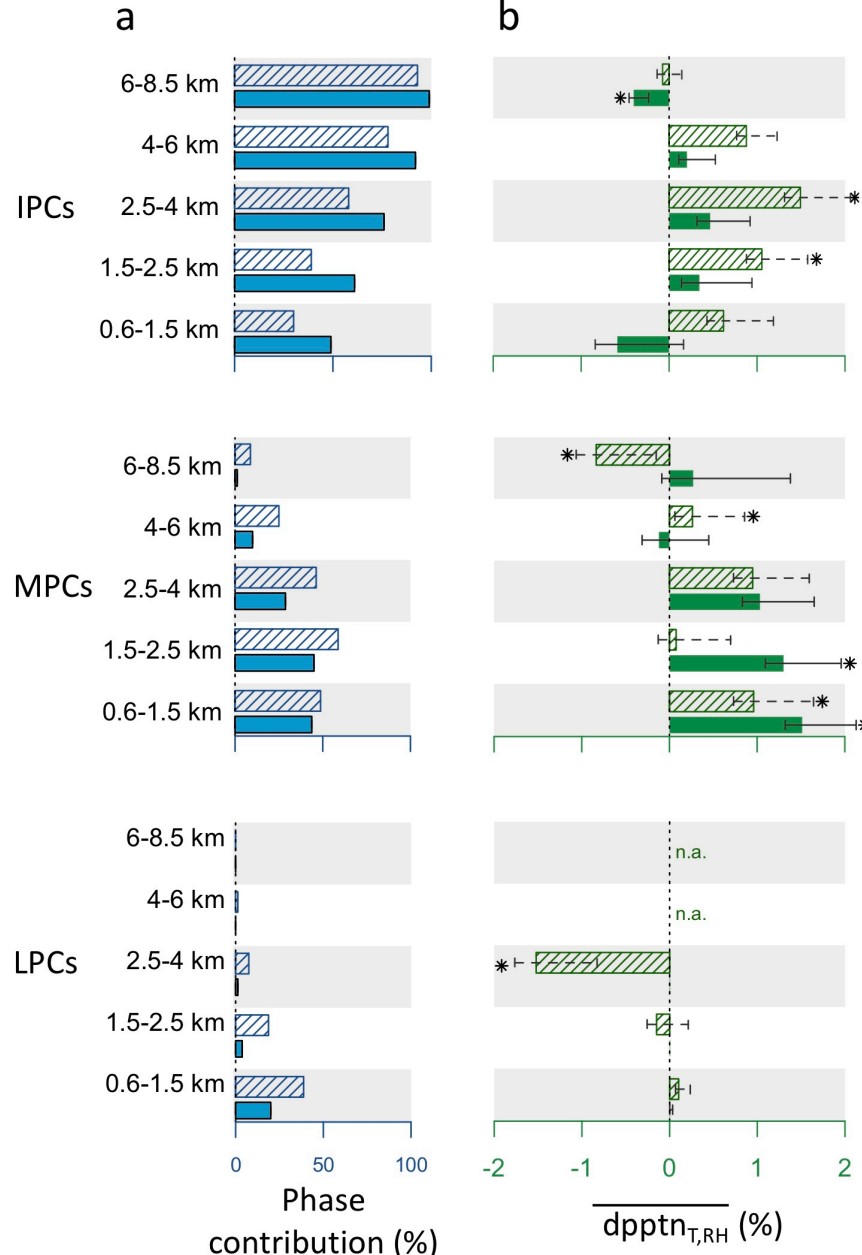

a

b

IPCs

MPCs

LPCs

Solid = Sea ice;   Hatched = Open ocean

**Figure 5: The absolute changes in a) CP distribution (blue), and b) $\overline{dpptn_{T,RH}}$ (green) for IPCs, MPCs, and LPCs over sea ice (solid) and open ocean (hatched) at different altitude ranges. An asterisk (*) indicates that the differences between all and clean conditions were significant for both relative and absolute values, based on a paired Wilcoxon rank test, p < 0.05, using T-RH grid cells containing > 800 (400 for $\overline{dCP_{T,RH}}$) 12.5-km$^2$ gridded observations. Error bars show bootstrapped 95% confidence intervals for the weighted mean.**