# Peer review of "A satellite-based estimate of combustion aerosol cloud microphysical effects over the Arctic Ocean"

_Atmospheric Chemistry and Physics, 2018_

## Referee Comment (RC1) · Anonymous Referee #1 · 9 Jul 2018

This paper uses estimates of cloud properties from satellite remote sensing (AIRS, CloudSat, CALIPSO, MODIS) and black carbon concentrations from the FLEXPART transport model to study cloud-aerosol microphysical effects over the Arctic Ocean. It is found that combustion aerosols are associated with large changes in surface longwave radiation over sea ice. However, up to 91% of the cloud fraction differences between all and clean conditions is due to meteorological conditions, i.e., the black carbon is essentially a passive tracer in these cases.

This paper is well-written and the results are very interesting. I think this paper is suitable for publication in ACP after addressing my concerns below.

Main comments:

[Figure]

1) There needs to be justification in the introduction for focusing on black carbon. It is not clear why other aerosol sources are excluded in this study. There needs to be an overview of previous studies of the role of black carbon in Arctic cloud-aerosol effects.

2) The measurements are separated into clear and cloudy conditions as a function of height. So different days are used in each of these averages? It would be very useful to have a figure showing what days where used in the different averages. Are the upper and lower quartiles used to make Figure 2 using data from different seasons? Are the averages as a function of height in Figure 1 using data from different seasons? If so, it obscures some of the effects since, for example, if the aerosols cause increased activation of ice crystals and precipitation at high altitudes then it will not be possible to see the impact of these at lower altitudes. I first interpreted Figure 2f as an example of increased ice production at higher altitudes and depletion of water vapor due to deposition at lower altitudes but this may not be the case if the values at different heights are calculated separately.

3) Throughout the paper it is said that this study is focused on regional-scale effects. What is meant by this exactly, that sea ice and open ocean is analyzed separately? For example, page 5, line 17, what is meant by "regionally averaged"?

4) The results indicating increased ice precipitation in MPC at low altitudes and decreased precipitation at high altitudes is very interesting. It would be good to include a more detailed comparison with the results from previous studies in the discussion section.

Minor comments:

1) Page 5, line 4: How does focusing on relative rather than absolute differences get around the issue of misclassification of small supercooled water as ice particles?

2) Page 7, line 21-22: This is a very interesting result but why wouldn't it be the case over open ocean?

3) Page 7-8, lines 33-1: Why is this relationship due to microphysical effects and not meteorology? Can it be concluded that microphysical effects are stronger in stable conditions or just more observable? If it is stronger then why is there no significant difference in the LTS for high and low quartile black carbon (Figure 2e)?

4) Page 8, lines 11-12: This is an interesting result but what would this be the case?

5) Page 10, lines 3-5: Are the results of this study consistent with the deactivation of pre-existing INP hypothesis (Archuleta et al. 2015; Cziczo et al. 2009)?

6) Page 11, lines 15-16: Why would the impact on MPCs be different over sea ice and open ocean?

---

## Referee Comment (RC2) · Anonymous Referee #2 · 2 Aug 2018

Review of

**A satellite-based estimate of aerosol-cloud microphysical effects over the Arctic Ocean**

Lauren M. Zamora,
Ralph A. Kahn, Klaus B. Huebert, Andreas Stohl, Sabine Eckhardt

**Summary**

The paper presents an analysis of aerosol-cloud effects using data from satellites and FLEXPART to provide first observation-based constraint on regional aerosol microphysical effects on total nighttime CF over the Arctic Ocean. An important feature is that the study accounts for the co-variation of airmass aerosol and meteorology. The topic and findings seem appropriate for the *ACP* readership.

Overall, the methodology seems sound and findings of interest. However, while the text is very well written from a grammatical standpoint, particularly in the presentation of the results the text is either disjointed or lacks sufficient information to follow in the figures/tables. Specific examples are given below for improving the flow of the text, along with other comments and question. The suggestions are not comprehensive so the authors are advised to please use them as a necessary-but-not-sufficient guide of how the manuscript should be modified for an outside reader to follow and absorb the content. For these reasons it is recommended that the paper be accepted but with major revision.

**Major Comments**

(Each specific comment is preceded by the page and line numbers. If in the supplemental material, only the line number is used.)

1) In the supplement, the evaluation of FLEXPART BC concentrations is based on CALIPSO aerosol profiles. Arguments are provided as to why BC would likely be the dominant aerosol type for the location and period of study. However, since CALIPSO cannot tell the difference between BC and other aerosol types, a more direct comparison would be if FLEXPART could provide the total distributions of all relevant aerosols for the region. Is that a capability of FLEXPART? If so, then FLEXPART could also be used to assess the fraction of the total aerosols that are BC and support the arguments provided.

2) P3: The focus is on BC concentrations, why? Please provide a justification, which appears to have been buried in the supplemental material.

3) In the results section, there is a lot of jumping back and forth between the figures in the main text and the supplemental material. As such, some of the supplemental material did not seem very "supplemental". Recommend moving frequently referred to figures or tables to the main text.

4) L94-100: The supplemental summary statement seems inconsistent. The last sentence states that "the model does represent aerosol transport over the Arctic well" but the first sentence states that "CALIPSO aerosol layers contributed significantly smaller volume than in all and

model-identified polluted conditions"; if the latter is correct, how can the former be? How does this affect the results? (particularly Fig. 4)

5) The issue with (4) might stem from confusion regarding Figure S3. The plot shows that fraction of the different altitude layers where CALIPSO detects aerosol and FLEXPART identifies (a) are clear, and (b) are polluted.

   a) While (a) are false negatives (consistent with the header at the top of the plot), is seems that (b) is inconsistent with its header and it is not false positives; rather it is showing when FLEXPART accurately identifies aerosol layers (i.e., CALIPSO=yes, and Pollution=yes). Is the caption wording correct?

   b) Also, the caption and the headers refer to "likely at large values" and "likely at small values"; values in what, BC concentration? If so, the plot only displays part of the information, the layer fractions and, besides the "clean" and "polluted" columns, there is not information on concentration level (that is consistent with the headers at the top of the figure). Please clarify.

6) Fig. 3: There are too many different aspects are loaded into this figure, making it very difficult to follow the discussed patterns in a single variable type with altitude (e.g., dCF). Recommend moving 3b to a new figure, and make a-c panel plots in Fig. 3 with altitude separately for (a) dCF, (b) dpptn, and (c) dCP(IPC, MPC, LPC). For the old 3b, the current overlays are too cluttered and recommend separating into (a) and (b) the pptn and CF components.

**Secondary Comments**

(Some rewordings are suggested that were easier for me to understand.)

7) P6, L25, "very large (~25 W m$^{-2}$)": Where does this value comes from? In the plots, values range from 0 to ~70 W m$^{-2}$. Please explain. In fact, more text is needed to explain Fig. 2 which is a 7 panel plot. Currently, it seems "dropped in" without many of its aspects discussed.

8) P7, L1, "up to 91% of the variability": Where is this value shown? I do not see any such value in Table S1.

9) P7, L11-13, "Cloud fraction substantially differed… At the lowest levels…": At the lowest level (0.6-1.5 km) over open ocean, almost all of the grids have Xs meaning that they are not statistically significant. Is it then a correct interpretation to say that they differed substantially?

10) P7, L20, "generally become more positive at constant RH with increasing T": This statement is not well supported given that many of the grids have white Xs preventing the "increasing with T" analysis.

11) Fig. 4: Please describe where the dots are from. Are they from the grids in the RH-T plots e.g. from Fig 1. but for each range of dBC? If so, are they only from those that are statistically different from zero?

12) P1, L19, "with implications for a warming Arctic." Such implications do not seem to have been discussed in the paper. Please add the discussion or remove this clause.

13) P2, L22, "Tropospheric cloud data…": Please indicate earlier/here the source here (CloudSat and CALIPSO); the details of the products can remain where they are.

14) P3, L26: "in general (Zamora et al. 2017)…" → "in general. Zamora et al. (2017)…"

15) P3, L27: What is meant by "strong aerosol layers"? Also, what does "aersosols[6]" mean?

16) P3, last line, "Cloud fraction is not well defined in the literature. Here, it is…": The statement is incorrect and unnecessary: CF is defined in the literature (tho its determination can be challenging). Recommend just starting off with "Cloud fraction is operationally…"

17) P4, L18, "and blowing snow": Your lowest altitude is 0.6 km; are you stating that blowing snow could be that high? If not, remove.

18) P4, L18-20, "Additionally, …": As stated, why is CloudSat mistaking precipitation for clouds an issue? If the lidar signal is attenuated it is attenuated and one has no signal to work with. This is true regardless of CloudSat's potential mistake. It would only factor in the precipitation counts, which does not seem to be the topic here.

19) P5, Section 2.3: Please include the local overpass time used from AIRS.

20) P6, L15, "Our focus on nighttime data over the flat ocean surface likely reduces effects from large-scale vertical motion": I do not know what you intend to mean by "large-scale vertical motion" since certainly large-scale synoptic phenomena exist at nighttime (fronts, highs, lows, etc.). Recommend rephrasing.

21) L17, What is the meaning of "convection[36]"?

22) Fig. 1: The white Xs indicate that the grid is not significantly different from zero. In the analyses that follow, are only the non-X grids used? Please state and/or give justification for inclusion if they are.

23) Fig. 3: Does the significance indicated by the asterisk apply to both the relative percent changes and absolute changes? Please state in text.

24) P7, L17, "1.7% to 0.7%": These values for sea ice do not match what I see, which is -2% to 1% (unless you maybe meant only at higher altitudes?).

25) P8, L1, "more influential at the lower temperatures": The values are near dCF=0 for the higher altitude points so is this an accurate statement, especially given the dramatic drop off over sea ice with lower altitude (warmer temperature) suggesting the dominance of the stability criterion?

26) P8, L6, "(Fig. 3)" → "(Fig. 3a)" for clarity.

27) P8, L11, "Over sea ice" → "From Table S2, over sea ice…" for clarity, as otherwise it was not clear what supported the last sentence in the paragraph.

28) P8, L18, "where LPC fractions were highest": Source for statement?

29) P8, L23, "An analysis…" → "We analyze the difference in precipitation frequency; however, an analysis…" (otherwise, the reader knows what you will not do, but it has not been stated what will be done).

30) P9, L1, "~91% of the MPCs": Shown where?

31) P8, L31, "night, potentially" → "night. This potentially leads to…" (break up the long sentence that also contains opposing points of view). And, which point of view does your study support?

32) P11, L12: Physically, why would one expect a larger aerosol effect for greater atmospheric stability?

33) P11, L25: Can you give a "for example" about what other cloud property relationships might exist?

34) Need the "author contributions" section for *ACP*

**Supplemental material**

35) L49-52: The concern about the variations in BC:OC ratios seems misplaced to me since the CALIPSO cannot tell the difference between those aerosol types.

36) L33-35, "it is unclear how thick an observed CALIPSO aerosol layer (measured in meters) must be to influence the average BC concentration in an altitude range…": The part "must be to influence the average BC concentration" seems odd in that there is no "influence" on the average BC concentrations. Please reword.

37) L38: "locations of false" → "locations of FLEXPART false"

38) L63-64: Please explain a bit more about how the FLEXPART and CALIPSO data are compared. Specifically, the text refers to percentages of a "layer volume"; is that to say that the CALIPSO layering is converted into a binary present/not present mask and compared to the equivalent binary from FLEXPART? If so, is there a criterion used for the binary CALIPSO masking?

39) L66: For clarity, recommend "so the fractions estimated" → "so the model false negative fractions estimated"

40) L69: Remove "However". It implies a course change from the prior text but one is not present.

41) L76: For clarity, "sea ice" → "sea ice (the pink line)"

42) L81: For clarity, recommend "detect non-dilute aerosol" → "detect (non-dilute) aerosol"

43) L82: "are likely to be"; shouldn't this be "will be" (?) since it is how you have defined false positives?

44) L86-87, "These aerosol…": I was looking for a figure to support the statement but it seems that one is not present? If there is, please indicate; if there isn't, please indicate "not shown."

45) Figure S4: Please rescale the y-axis to cover the range of the bars plotted (i.e., most "polluted" bars exceed the plot range).

46) Table S2: The values in the square brackets are defined in the caption, but what are the values in the parentheses that precede the square brackets? Recommend rearranging to be in column format as the font size to fit in portrait is too small to read easily.

---

## Author Comment (AC1) · 11 Sep 2018

**Response to Referee #1:**

This paper uses estimates of cloud properties from satellite remote sensing (AIRS, CloudSat, CALIPSO, MODIS) and black carbon concentrations from the FLEXPART transport model to study cloud-aerosol microphysical effects over the Arctic Ocean. It is found that combustion aerosols are associated with large changes in surface longwave radiation over sea ice. However, up to 91% of the cloud fraction differences between all and clean conditions is due to meteorological conditions, i.e., the black carbon is essentially a passive tracer in these cases.

This paper is well-written and the results are very interesting. I think this paper is suitable for publication in ACP after addressing my concerns below.

*We kindly thank this Referee for their very helpful comments. Please find our responses below. Bold lettering indicates newly added text.*

Main comments:
1) There needs to be justification in the introduction for focusing on black carbon. It is not clear why other aerosol sources are excluded in this study. There needs to be an overview of previous studies of the role of black carbon in Arctic cloud-aerosol effects.

*We have added the following paragraph to the introduction to a) address why the study focused on combustion aerosols (as proxied by BC), instead of on other aerosol types, and b) present an overview of previous studies relevant to combustion microphysical effects on clouds:*

"**In this paper, we focus on the effects of combustion-derived (i.e., anthropogenic pollution plus smoke) aerosols on clouds. Combustion-derived aerosols are strongly impacted by anthropogenic activity, and tend to dominate columnar mass under high AOD conditions in the Arctic (Xie et al., 2018), although in spring the more well-mixed mineral dust can also contribute ~10% to total Arctic AOD levels (Breider et al., 2014; Groot Zwaaftink et al., 2016). Combustion aerosols have pronounced effects on Arctic cloud microphysical and radiative properties (e.g., Carrió et al. (2005); Coopman et al. (2016, 2018); Earle et al. (2011); Garrett et al. (2004); Jouan et al. (2014); Lubin and Vogelmann (2006); Tietze et al. (2011); Zamora et al. (2016, 2017); Zhao and Garrett (2015)). Their cloud impacts are likely to be particularly large during winter and spring, when they are transported to the Arctic most efficiently, and when precipitation is reduced, causing a peak in aerosol abundance at many remote Arctic ground stations known as "Arctic Haze" (Barrie, 1986; Croft et al., 2016; Quinn et al., 2007; Stohl, 2006). However, so far it has been challenging to assess their cloud effects on the Arctic region as a whole, due to large cloud model uncertainties, spatial/temporal observation limitations, and difficulties obtaining some remote sensing information at high latitudes.**"

*We also added a clarification in section 2.1, where we explained why BC specifically is used as a proxy for combustion aerosols:*

"FLEXPART BC is used in this study as a proxy for all combustion aerosols, **because they very often contain BC, although in somewhat different fractions**. The association of high levels of modeled BC with CALIPSO aerosols in general (see Zamora et al. (2017)) indicates that modeled BC is a fairly good proxy for strong (CALIPSO-detectable) aerosol layers during polar night, even though some local sources of combustion aerosols (Creamean et al., 2018; Maahn et al., 2017) might not be included in the model. Model comparisons to CALIPSO aerosol data in the study region also indicate that model-identified clean conditions (BC $< 30$ ng m$^{-3}$) are associated with significantly lower levels of CALIPSO aerosol layer presence relative to average or polluted conditions (see supplement for further details)."

2) The measurements are separated into clear and cloudy conditions as a function of height. So different days are used in each of these averages?

*To clarify, we did not separate measurements into clear and cloudy conditions for the main body of work described in the paper. We identified the subset of the data present in air mass conditions determined to be clean. This subset was comprised of a reduced number of days relative to the full dataset. Because there are many clean cases, and because defining a lower bound on "polluted" cases would introduce an arbitrary threshold, we compare the clean subset to the full dataset, which includes the clean subset.*

It would be very useful to have a figure showing what days were used in the different averages.

*We now show what days were used in the different averages in the new Table 2. We also added some associated discussion below:*

"To help better understand co-varying meteorological effects on CF specifically, we assessed a generalized additive model (GAM) (Hastie and Tibshirani, 1990) of the $\overline{dRH}$, $\overline{dT}$, and $\overline{dCF}$ data at each vertical level, season, and surface type (Table 1). **Seasonal differences in light, sea ice extent, and BC levels led to some sample number differences for sea ice and open ocean at different times of the year (Table 2). In the GAM, the seasonal values in Table 1 were weighted equally to represent the equal periods of the year being sampled.**"

Are the averages as a function of height in Figure 1 using data from different seasons?

*Yes, the averages in Figure 1 are calculated from all seasons. We now state this in the text.*

If so, it obscures some of the effects since, for example, if the aerosols cause increased activation of ice crystals and precipitation at high altitudes then it will not be possible to see the impact of these at lower altitudes.

*That is a good point. To address this concern, we now show the data separated by season (see the new Figures S1-S3). Separating by season results in some loss of sample sizes in the individual figures, and reduced statistical significance for the individual seasonal plots. However, where sample sizes were large enough, the seasonal data show very similar vertical trends in dCF as during the annual average shown in Figure 1. To draw attention to this new information, we have added the following text in the main document:*

*"*Cloud fraction substantially differed among all and clean conditions for many combinations of T, RH, altitude and surface type (Fig. 1). Estimated aerosol impacts on total CF depend on altitude and surface type**, but are fairly consistent among seasons (Figs. S1-S3)**."

Are the upper and lower quartiles used to make Figure 2 using data from different seasons? I first interpreted Figure 2f as an example of increased ice production at higher altitudes and depletion of water vapor due to deposition at lower altitudes, but this may not be the case if the values at different heights are calculated separately.

*We now clarify in the text that Figure 2 does include data from all seasons, and that Fig. 2f and 2g represent median profiles (and thus the T and RH profiles were not calculated separately at different heights, unlike in Figure 1).*

"To illustrate this point, Figure 2 shows the longwave $CRE_{BOA}$ for the upper and lower quartiles of FLEXPART model column BC concentrations**, calculated** during the **entire** study period. The upper and lower quartile ranges of column BC levels are associated with very large (~10 W m$^{-2}$) differences in **median** longwave $CRE_{BOA}$ over sea ice (Fig. 2). However, **when we compare the median relative humidity and temperature profiles with column BC levels in the upper quartile over sea ice (Fig. 2f, red lines) and open ocean (Fig. 2g, red lines) to the lower quartile profiles (blue lines, same figures), it is clear that** column BC levels over sea ice are also associated with noticeable differences in **median** relative humidity and temperature **profiles** (Fig. 2f). Small differences in lower tropospheric stability (Fig. 2e), defined as the difference in potential temperature between 700 and 1000 hPa, are also observed. These meteorological factors strongly affect CF and CP, which in turn help drive $CRE_{BOA}$. As a result, aerosol microphysical effects may contribute only a fraction of the $CRE_{BOA}$ differences shown in Figure 2."

*More investigation would be required to verify the referee's original interpretation about increased ice production. Such work is beyond the scope of this manuscript, but would certainly be interesting to pursue further.*

3) Throughout the paper it is said that this study is focused on regional-scale effects. What is meant by this exactly, that sea ice and open ocean is analyzed separately? For example, page 5, line 17, what is meant by "regionally averaged"?

*To avoid confusion, we have changed this wording throughout the paper. For example, see the edited specific text the referee pointed out above:*

" **D**ifferences in relative humidity, temperature, and 12.5 km$^2$ gridded CF ($\overline{dRH}$, $\overline{dT}$, and $\overline{dCF}$, respectively) between all ($\overline{RH}$, $\overline{T}$, and $\overline{CF}$, respectively) and clean ($\overline{RH_c}$, $\overline{T_c}$, and $\overline{CF_c}$, respectively) conditions were calculated **over sea ice and open ocean regions**…"

4) The results indicating increased ice precipitation in MPC at low altitudes and decreased precipitation at high altitudes is very interesting. It would be good to include a more detailed comparison with the results from previous studies in the discussion section.

*Thank you. We have added the following new text in the Introduction:*

[revised manuscript text omitted]

Minor comments:

1) Page 5, line 4: How does focusing on relative rather than absolute differences get around the issue of misclassification of small supercooled water as ice particles?

*We have re-worded for clarification:*

"Previous work shows that this product can severely underestimate downwelling LW radiation due to misclassification of small supercooled water as ice particles (Van Tricht et al., 2016)**, leading to uncertainties in the absolute values of $CRE_{BOA}$. Here**, we primarily focus on **relative differences** in $CRE_{BOA}$ **between two subsets of data: those with high and low modeled BC values. The uncertainty due to misclassification of small particle phase is similar in both subsets of data, which are collected over the same surfaces and years, allowing for meaningful comparisons to be made between the two datasets despite uncertainty in the absolute values.**"

2) Page 7, line 21-22: This is a very interesting result but why wouldn't it be the case over open ocean?

*Based on comments from the other referee, we have taken this paragraph out, as it was not fully substantiated.*

3) Page 7-8, lines 33-1: Why is this relationship due to microphysical effects and not meteorology? Can it be concluded that microphysical effects are stronger in stable conditions or just more observable? If it is stronger then why is there no significant difference in the LTS for high and low quartile black carbon (Figure 2e)?

*Thanks for pointing this out. We have re-worded to make it clearer that the microphysical effects are more apparent under certain meteorological conditions, where their effects are less likely to be overwhelmed by other factors:*

"This finding suggests that aerosol microphysical impacts on low-altitude clouds are more **observable** at the lower temperatures and/or more stable conditions over sea ice. Previous studies have also observed **more apparent** aerosol microphysical effects under more stable conditions in the Arctic (Coopman et al., 2018; Zamora et al., 2017)."

4) Page 8, lines 11-12: This is an interesting result but what would this be the case?

*We have changed the text as follows:*

"From Table S2, over sea ice between 1.5-2.5 km, the relative contributions of LPCs and MPCs were significantly lower at high $dBC_{T,RH}$ levels (>20 ng m$^{-3}$), whereas that of IPCs was significantly higher. **The reduction in liquid-containing clouds at higher $dBC_{T,RH}$ levels is consistent with a glaciation effect, whereby increased presence of aerosols leads to ice particle formation and cloud dissipation, as observed in section 3.1.**"

5) Page 10, lines 3-5: Are the results of this study consistent with the deactivation of pre-existing INP hypothesis (Archuleta et al. 2015; Cziczo et al. 2009)?

*We have changed the text, as follows:*

"Alternatively, combustion aerosols might reduce the efficiency of pre-existing INPs through the "deactivation effect" (Archuleta et al., 2005; Cziczo et al., 2009). Reduced ice crystal formation rates could then lead to more frequent air mass saturation with respect to liquid water, more water droplets that freeze homogeneously, and smaller, more numerous ice particles, and less precipitation (Girard et al., 2013) **as observed here**. This effect could lead to enhanced total CF over the Arctic (Du et al., 2011). Although absolute humidity within the different T-RH bins between 6-8.5 km is not systematically related to higher $dBC_{T,RH}$ levels as one might expect with the deactivation effect, it is possible that pre-sorting the data by 5% RH bins to reduce the impacts of meteorological co-variation could make evidence for this effect more difficult to observe.

**Therefore, it is difficult to say whether this study is consistent with the deactivation effect hypothesis, but it does not preclude it.**"

6) Page 11, lines 15-16: Why would the impact on MPCs be different over sea ice and open ocean?

*We took out the "relative to IPCs" text in the paragraph of reference below, since the difference relative to IPCs was not significant (see Fig. 3a), and have added the following text:*

"Below 1.5 km, we also observed a 7% reduction in the LPC and MPC fractions over sea ice, but a slight increase in MPCs relative to  LPCs over open ocean. **The different effect on MPCs over sea ice and open ocean may be related to the higher temperatures over open ocean, leading to less efficient ice formation, or to some other, as yet unknown, factor.**"

**Response to Referee #2:**

Summary

The paper presents an analysis of aerosol-cloud effects using data from satellites and FLEXPART to provide first observation-based constraint on regional aerosol microphysical effects on total nighttime CF over the Arctic Ocean. An important feature is that the study accounts for the co-variation of airmass aerosol and meteorology. The topic and findings seem appropriate for the ACP readership.

Overall, the methodology seems sound and findings of interest. However, while the text is very well written from a grammatical standpoint, particularly in the presentation of the results the text is either disjointed or lacks sufficient information to follow in the figures/tables. Specific examples are given below for improving the flow of the text, along with other comments and question. The suggestions are not comprehensive so the authors are advised to please use them as a necessary-but-not-sufficient guide of how the manuscript should be modified for an outside reader to follow and absorb the content. For these reasons it is recommended that the paper be accepted but with major revision.

*We appreciate the time the referee took to give such helpful and thorough comments, which have improved the quality and clarity of the manuscript. We particularly appreciate the reviewer pointing out cases where other reader might have found things confusing. Please find our responses below. Bold lettering indicates newly added text.*

Major Comments

(Each specific comment is preceded by the page and line numbers. If in the supplemental material, only the line number is used).

1. In the supplement, the evaluation of FLEXPART BC concentrations is based on CALIPSO aerosol profiles. Arguments are provided as to why BC would likely be the dominant aerosol type for the location and period of study. However, since CALIPSO cannot tell the difference between BC and other aerosol types, a more direct comparison would be if FLEXPART could provide the total distributions of all relevant aerosols for the region. Is that a capability of FLEXPART? If so, then FLEXPART could also be used to assess the fraction of the total aerosols that are BC and support the arguments provided.

*This is a good suggestion. However, FLEXPART is no "normal" aerosol model but rather a Lagrangian tracer model. That means that simulations are normally done for one specific aerosol type and not for a full suite of different aerosols, as is often available in other models. The advantage of FLEXPART is its higher accuracy with respect to long-range transport (notice that "normal" Eulerian transport models suffer from excessive numerical dissipation, failing to preserve chemical plumes over long transport distances, see, e.g., Eastham and Jacob (2017)). FLEXPART's disadvantage is that it is not built for simulating many species at the same time, including their interactions. For this study, we think it is more important to have the plumes of a "representative" combustion aerosol species at the right place at the right time, rather than to quantify the contributions of many different species.*

*Dust is definitely another important aerosol type in the Arctic, and is not well correlated with BC. Limited simulations of dust have been done with FLEXPART (Groot Zwaaftink et al., 2016). It was found that local "sharp" plumes are mainly present in the fall, whereas dust emissions in winter are limited by extensive snow cover in the Arctic. On the other hand, long-range transport of dust from low-latitude sources (e.g., from the Sahara) also occurs in winter. However, dust from these very remote sources is well mixed and, at moderate concentrations, nearly omni-present in the Arctic free troposphere, so that it would not be straightforward to distinguish "dusty" and "clean" air masses.*

*Based on this information, we have addressed the referee's comment above in four ways. First, we have added supporting information on dust aerosol sources:*

"Mineral dust can be found throughout the Arctic atmosphere. **However, although there are some local "sharp" dust plumes at some locations in the fall, wintertime local dust emissions are limited by extensive snow cover, and long-range transport of low-latitude dust is well mixed in the winter and, at moderate concentrations, is nearly omni-present in the Arctic free troposphere (Breider et al., 2014; Groot Zwaaftink et al., 2016).**"

*Second, we have clarified our focus better by changing the title:*

"A satellite-based estimate of **combustion** aerosol cloud microphysical effects over the Arctic Ocean"

*And we make it clearer throughout the text that we are focusing on the specific microphysical effects of combustion aerosols on clouds in this study, and not on the effects of all aerosols on clouds.*

*Thirdly, we better clarify how to interpret the information in the supplement. We make it clearer that our main focus in the supplement is on identifying:*

1) ***false positives*** *(where BC aerosols were not present as evidenced by the lack of a CALIPSO aerosol layer, but the model said they were), and*
2) ***an upper limit on false negatives*** *(where BC aerosols could have been present based on the presence of a CALIPSO layer, but the model said they were not).*

*Note that based on our method, additional information on the distributions of other aerosol types from FLEXPART would not affect the false positive rate. It is possible that this information could help refine and lower the estimated potential false negative rate. However, because FLEXPART dust aerosols are poorly validated, especially over oceanic regions of the Arctic, we would be adding in an unknown level of uncertainty in the false negative estimates if we assume that the poorly validated distributions of dust are correct, and we use these estimates of other aerosols to lower the estimated false negative rate for combustion aerosols. Under our current assumptions, we are only able to provide an upper estimate of the false negative rate, but this upper limit is fairly well-constrained.*

*Lastly, we would like to point out that although this work adds additional validation of FLEXPART BC levels by comparison to CALIPSO aerosol profiles, it is by no means the only validation of FLEXPART Arctic BC. Previous validation, with positive results, has been based on aircraft and ground measurements, and other satellite data. We cite some of this previous work in section 2.1:*

"FLEXPART is widely used, and is well-validated for the purpose of studying Arctic smoke and pollution transport (Damoah et al., 2004; Eckhardt et al., 2015; Forster et al., 2001; Paris et al., 2009; Sodemann et al., 2011; Stohl et al., 2002, 2003, 2015; Zamora et al., 2017)."

*Because the information in the supplementary material provides better constraints on FLEXPART BC vertical distributions over open ocean, we think it is a useful contribution. However, the previous studies cited above add much more confidence to our ability to correctly identify combustion aerosol layers with the model than the information provided in the supplement alone. Note also that there are a variety of other previous related studies that have used FLEXPART to identify combustion aerosols for aerosol-cloud interaction work, and these studies did not have the benefit of the extra*

*knowledge we provide in the supplementary material (e.g., Coopman et al. (2016, 2018); Tietze et al. (2011); Zamora et al. (2017)). Therefore, even though there are some uncertainties in our analysis in the supplementary material, we do not think they would preclude the greater work from being useful.*

2. P3: The focus is on BC concentrations, why? Please provide a justification, which appears to have been buried in the supplemental material.

*We have added the following paragraph to the introduction to address why the study focused on combustion aerosols (as proxied by BC), instead of on other aerosol sources:*

"**In this paper, we focus on the effects of combustion-derived (i.e., anthropogenic pollution plus smoke) aerosols on clouds. Combustion-derived aerosols are strongly impacted by anthropogenic activity, and tend to dominate columnar mass under high AOD conditions in the Arctic (Xie et al., 2018), although in spring the more well-mixed mineral dust can also contribute ~10% to total Arctic AOD levels (Breider et al., 2014; Groot Zwaaftink et al., 2016). Combustion aerosols have pronounced effects on Arctic cloud microphysical and radiative properties (e.g., Carrió et al. (2005); Coopman et al. (2016, 2018); Earle et al. (2011); Garrett et al. (2004); Jouan et al. (2014); Lubin and Vogelmann (2006); Tietze et al. (2011); Zamora et al. (2016, 2017); Zhao and Garrett (2015)). Their cloud impacts are likely to be particularly large during winter and spring, when they are transported to the Arctic most efficiently, and when precipitation is reduced, causing a peak in aerosol abundance at many remote Arctic ground stations known as "Arctic Haze" (Barrie, 1986; Croft et al., 2016; Quinn et al., 2007; Stohl, 2006). However, so far it has been challenging to assess their cloud effects on the Arctic region as a whole, due to large cloud model uncertainties, spatial/temporal observation limitations, and difficulties obtaining some remote sensing information at high latitudes.**"

*We also added a clarification in section 2.1, where we explained why BC specifically is used as a proxy for combustion aerosols:*

"FLEXPART BC is used in this study as a proxy for all combustion aerosols, **because they very often contain BC, although in somewhat different fractions**. The association of high levels of modeled BC with CALIPSO aerosols in general (see Zamora et al. (2017)) indicates that modeled BC is a fairly good proxy for strong (CALIPSO-detectable) aerosol layers during polar night, even though some local sources of combustion aerosols (Creamean et al., 2018; Maahn et al., 2017) might not be included in the model. Model comparisons to CALIPSO aerosol data in the study region also indicate that model-identified clean conditions (BC < 30 ng m$^{-3}$) are associated with significantly lower levels of CALIPSO aerosol layer presence relative to average or polluted conditions (see supplement for further details)."

3. In the results section, there is a lot of jumping back and forth between the figures in the main text and the supplemental material. As such, some of the supplemental material did not

seem very "supplemental". Recommend moving frequently referred to figures or tables to the main text.

> *The original Table S2 was the most referenced supplemental item. This table presents the same information as in the original Figure 3, except that it also shows which cases were significantly different at high (> 20 ng m$^{-3}$) BC levels. We have now added this information to the new Figure 3 (as the orange triangles), and are thus able to remove the references to Table S2 in the main text. Table S1 was referenced twice, and has been moved to the main text. Figure S1 and S2 are now only referenced once each, and remain in the supplement.*

4.  L94-100: The supplemental summary statement seems inconsistent. The last sentence states that "the model does represent aerosol transport over the Arctic well" but the first sentence states that "CALIPSO aerosol layers contributed significantly smaller volume than in all and model-identified polluted conditions"; if the latter is correct, how can the former be? How does this affect the results? (particularly Fig. 4)

> *Our apologies for the confusing wording that led the referee to wonder if there was an inconsistency. A low aerosol volume in clean conditions is what one would expect if the model was performing well. With re-wording of this paragraph, we hope that it is now clear that there is no inconsistency:*
>
> *"**In summary, for FLEXPART to correctly identify clean (i.e., low combustion aerosol) conditions, it needs to be able to correctly simulate the horizontal and vertical distributions of combustion aerosols**. Previously conducted model validation studies indicate that FLEXPART has skill in simulating the horizontal locations of BC transport over the Arctic. Here, we show that the volume of CALIPSO vertical aerosol layers is significantly smaller in model-estimated clean conditions in the vertical column than in all conditions, or in** model-identified polluted conditions. **This result indicates that FLEXPART also has some skill in the vertical layer prediction of BC aerosols over the Arctic Ocean. Moreover, we observed** no major spatial biases in the false negative rates that would preclude the regional comparisons between sea ice and open ocean regions. **Together, these findings and previous work support the use of FLEXPART for identifying clean conditions for the purposes of this study.**"*

5.  The issue with (4) might stem from confusion regarding Figure S3. The plot shows that fraction of the different altitude layers where CALIPSO detects aerosol and FLEXPART identifies (a) are clear, and (b) are polluted.

a)  While (a) are false negatives (consistent with the header at the top of the plot), is seems that (b) is inconsistent with its header and it is not false positives; rather it is showing when FLEXPART accurately identifies aerosol layers (i.e., CALIPSO=yes, and Pollution=yes). Is the caption wording correct?

*To be more accurate, we would slightly rephrase the referee's interpretation above. In Figure S3b we have plotted only the subset of cases with high BC levels (Pollution = yes). Figure S3b does show when FLEXPART is likely to have accurately identified aerosol layers (when the color axis values are closer to red, CALIPSO=yes). Red colors indicate that the model vertical layer was, on average, associated with a large CALIPSO aerosol layer at that location.*

*However, the same figure also shows the opposite (when values are closer to white, CALIPSO=no), indicating that the model vertical layer was, on average, not associated with a large CALIPSO aerosol layer at that location. With the header, we had intended to guide the reader to focus on those values that were closer to white: i.e., when false negatives are more likely to be present. We have now re-worded the caption to hopefully make this clearer.*

b)  Also, the caption and the headers refer to "likely at large values" and "likely at small values"; values in what, BC concentration? If so, the plot only displays part of the information, the layer fractions and, besides the "clean" and "polluted" columns, there is not information on concentration level (that is consistent with the headers at the top of the figure). Please clarify.

*We were referring to values in the color axis, which show to the fraction of model vertical layer containing an observed CALIPSO aerosol layer averaged at that location. To interpret the plot, no additional BC concentration information is needed, other than the knowledge that Figure S3a only shows the clean subset (all data with BC < 30 ng m$^{-3}$), and Figure S3b only shows the polluted subset (all data with BC >150 ng m$^{-3}$). This information has now been better clarified in the caption.*

6.   Fig. 3: There are too many different aspects are loaded into this figure, making it very difficult to follow the discussed patterns in a single variable type with altitude (e.g., dCF). Recommend moving 3b to a new figure, and make a-c panel plots in Fig. 3 with altitude separately for (a) dCF, (b) dpptn, and (c) dCP(IPC, MPC, LPC). For the old 3b, the current overlays are too cluttered and recommend separating into (a) and (b) the pptn and CF components.

*We have followed the referee's suggestions (see the new Figures 3 and 5). Note that there were two errors in the old Figure 3a. First, the labeling for the relative and absolute columns were reversed (this is now fixed in the new Figure 3). Also, we have corrected an error in the confidence interval values, which are now substantially reduced in most cases.*

Secondary Comments

(Some rewordings are suggested that were easier for me to understand.)

7. P6 , L25,"very large (~25 W m$^{-2}$)": Where does this value come from? In the plots, values range from 0 to ~70 W m$^{-2}$. Please explain.

*We are glad you asked us to clarify our methods here, because in doing so, we found an error in how these values were calculated. The corrected median CRE$_{BOA}$ difference over sea ice is now reduced to 10 W m$^{-2}$ from 25 W m$^{-2}$. The general conclusions remain the same though, since this is still quite a large difference.*

*To clarify our methods, we have added the following text:*

"The upper and lower quartile ranges of column BC levels are associated with very large (~**10** W m$^{-2}$) differences in longwave CRE$_{BOA}$ over sea ice (Fig. 2)**. This value is estimated from the median difference in 12.5 km$^2$ gridded CRE$_{BOA}$ values over sea ice regions across the Arctic Ocean during the study period, in grid cells with a minimum of at least 3 observations in the upper and lower quartile ranges of column BC levels**."

In fact, more text is needed to explain Fig. 2 which is a 7-panel plot. Currently, it seems "dropped in" without many of its aspects discussed.

*In addition to the above text, we have also added in the following text:*

"Systematic regional co-variability of aerosols and meteorological factors must be accounted for in order to avoid overestimating aerosol impacts on clouds (Coopman et al., 2018; Gryspeerdt et al., 2016). To illustrate this point, Figure 2 shows the longwave CRE$_{BOA}$ for the upper and lower quartiles of FLEXPART model column BC concentrations**, calculated** during the **entire** study period. The upper and lower quartile ranges of column BC levels are associated with very large (~**10** W m$^{-2}$) differences in **median** longwave CRE$_{BOA}$ over sea ice (Fig. 2). **However, when we compare the median relative humidity and temperature profiles with column BC levels in the upper quartile over sea ice (Fig. 2f, red lines) and open ocean (Fig. 2g, red lines) to the lower quartile profiles (blue lines, same figures), it is clear that** column BC levels **over sea ice** are also associated with noticeable differences in **median** relative humidity **and temperature profiles (Fig. 2f). Small differences in lower tropospheric stability (Fig. 2e), defined as the difference in potential temperature between 700 and 1000 hPa, are also observed.** These meteorological factors strongly affect CF and CP, which in turn help drive CRE$_{BOA}$. As a result, aerosol microphysical effects may contribute to only a fraction of the CRE$_{BOA}$ differences shown in Figure 2."

*And in section 3.1:*

"Also, dCF$_{T,RH}$ changes at high dBC$_{T,RH}$/low altitude are **more observable** over sea ice (Fig. 4)**, where lower tropospheric stability was greater and temperatures were colder (Figs. 2e-g)**."

8.  P7, L1, "up to 91% of the variability": Where is this value shown? I do not see any such value in Table S1.

*This GAM result is a single value calculated from the data in Table S1 (now Table 1), and thus does not easily fit in that table or in any of the other figures or tables. Therefore, it is only presented in the text. To help clarify our methods regarding the GAM calculations, we have added the following text:*

"To help better understand co-varying meteorological effects on CF specifically, we assessed a generalized additive model (GAM) (Hastie and Tibshirani, 1990) of the $\overline{dRH}$, $\overline{dT}$, and $\overline{dCF}$ data at each vertical level, season, and surface type (Table 1). **Seasonal differences in light, sea ice extent, and BC levels led to some sample number differences for sea ice and open ocean at different times of the year (Table 2). In the GAM, the seasonal values in Table 1 were weighted equally to represent the equal periods of the year being sampled.**

The GAM suggests that co-varying differences in $\overline{dRH}$ and $\overline{dT}$ by themselves can explain up to 91% of the variability in $\overline{dCF}$ (as measured by deviance, a statistic similar to variance (Jorgensen, 1997)). Because aerosols can co-vary with T and RH (e.g., because polluted air masses are more likely to have recently resided near the continental surface than clean air masses), aerosols could be responsible for some of this explained variability even without being explicitly included in this GAM. **For reference, a GAM based only on $\overline{dBC}$ explained up to 40% of $\overline{dCF}$ variability.** Thus, the 91% value is an upper estimate of the $\overline{dRH}$ and $\overline{dT}$ influences. Nonetheless, these findings underscore the importance of interpreting aerosol effects on clouds in the context of co-varying temperature and relative humidity. They also indicate that changes in T and RH of air masses entering the Arctic will likely have a very large influence on observed CF, to a degree that is likely to be much more regionally important than the microphysical effects of the aerosols themselves."

9.  P7, L11-13, "Cloud fraction substantially differed...At the lowest levels...": At the lowest level (0.6-1.5 km) over open ocean, almost all of the grids have Xs meaning that they are not statistically significant. Is it then a correct interpretation to say that they differed substantially?

*The text referred to above is:*

"Cloud fraction substantially differed among all and clean conditions for many combinations of T, RH, altitude and surface type (Fig. 1). Estimated aerosol impacts on total CF depend on altitude and surface type. At the lowest levels (0.6-2.5 km over sea ice and 0.6-1.5 km over open ocean), weighted mean dCF$_{T,RH}$ ($\overline{dCF_{T,RH}}$) is negative, resulting in an ~6% reduction in CF relative to clean conditions over sea ice (-0.6% over open ocean) (Fig. 3)."

*We believe the statement above is correct. From the above text, note that we do not say that there were substantial differences in all locations and at all individual T-RH grid*

*cells. We state that many of the grid cells differ substantially, but that the impacts depended on altitude and surface type. We also stated that the weighted mean is significantly different among all grid cells (also see our response to question 22 below).*

10. P7, L20, "generally become more positive at constant RH with increasing T": This statement is not well supported given that many of the grids have white Xs preventing the "increasing with T" analysis.

*To address this concern, we have taken this statement out.*

11. Fig. 4: Please describe where the dots are from. Are they from the grids in the RH-T plots e.g. from Fig 1. but for each range of dBC? If so, are they only from those that are statistically different from zero?

*We now describe this better in the Figure caption: "**In order to avoid obscuring emergent properties of the full dataset, the data include all meteorological conditions, including those where $dCF_{T,RH}$ are not significantly different from zero (as noted by white Xs in Figure 1)**."*

12. P1, L19, "with implications for a warming Arctic." Such implications do not seem to have been discussed in the paper. Please add the discussion or remove this clause.

*To better support this statement, we now add the following information in the "Summary and conclusions" section:*

"Observations from others (e.g., Chernokulsky et al. (2017); Eastman and Warren (2010)) show that expansion of open ocean areas appears to be connected to changing Arctic Ocean cloud properties. The different cloud responses to aerosols that we observe over sea ice vs. open ocean may provide partial clues into the cause of this behaviour, and into the future impacts of combustion aerosols on the Arctic system in general."

13. P2, L22, "Tropospheric cloud data...": Please indicate earlier/here the source here (CloudSat and CALIPSO); the details of the products can remain where they are.

*We have added this information, as suggested.*

14. P3, L26: "in general (Zamora et al. 2017)..." → "in general. Zamora et al. (2017)..."

*To clarify, the sentence now reads:*

"FLEXPART BC is used in this study as a proxy for all combustion aerosols. **T**he association of high levels of modeled BC with CALIPSO aerosols in general **(see** Zamora et al. (2017)**)** indicates that modeled BC is a fairly good proxy for strong (CALIPSO-detectable) aerosol layers during polar night, even though some local sources of

combustion aerosols (Creamean et al., 2018; Maahn et al., 2017) might not be included in the model."

15. P3, L27: What is meant by "strong aerosol layers"? Also, what does "aersosols[6]" mean?

*To clarify, we now state* "…strong **(CALIPSO-detectable)**". "aersosols[6]" *was a typo. It now reads:* "aerosols **(Creamean et al., 2018; Maahn et al., 2017)**"

16. P3, last line, "Cloud fraction is not well defined in the literature. Here, it is…": The statement is incorrect and unnecessary: CF is defined in the literature (tho its determination can be challenging). Recommend just starting off with "Cloud fraction is operationally…"

*Edited as suggested.*

17. P4, L18, "and blowing snow": Your lowest altitude is 0.6 km; are you stating that blowing snow could be that high? If not, remove.

*We spoke with a blowing snow expert (Y. Yang, personal communication), and were told that blowing snow up to 0.6 km in the Arctic above sea ice is not common, but is possible.*

18. P4, L18-20, "Additionally, …": As stated, why is CloudSat mistaking precipitation for clouds an issue? If the lidar signal is attenuated it is attenuated and one has no signal to work with. This is true regardless of CloudSat's potential mistake. It would only factor in the precipitation counts, which does not seem to be the topic here.

*To clarify, if CloudSat wrongly misclassifies precipitation as a cloud, that not only adds error into the precipitation estimates, but also to the CF estimates that are key to this work. When available, CALIPSO lidar data add additional information to help reduce the probability of this misclassification by the CloudSat radar, but these data are not available under optically thick clouds.*

19. P5, Section 2.3: Please include the local overpass time used from AIRS.

*This information is now added.*

20. P6, L15, "Our focus on nighttime data over the flat ocean surface likely reduces effects from large-scale vertical motion": I do not know what you intend to mean by "large-scale vertical motion" since certainly large-scale synoptic phenomena exist at nighttime (fronts, highs, lows, etc.). Recommend rephrasing.

*Thank you, this now reads* "Our focus on nighttime data over the flat ocean surface likely reduces effects from **solar-heating-driven** vertical motion"

21. L17, What is the meaning of "convection[36]"?

*That was a typo. It now reads:* "convection **(Serreze and Barry, 2005)**"

22. Fig. 1: The white Xs indicate that the grid is not significantly different from zero. In the analyses that follow, are only the non-X grids used? Please state and/or give justification for inclusion if they are.

*We include all the data in the analyses, including those in gridcells marked by Xs in Figure 1. There are both technical and scientific reasons for doing so.*

*To clarify the technical reason (i.e., that the important thing about the white Xs in Figure 1 is their number and not their individual positions), we have added the following text to the caption of Figure 1:*

"Figure 1: An example of $dCF_{T,RH}$ output at each altitude level. For illustration purposes, here each grid cell represents $\geq 7500$ km$^2$ of gridded observations. Blue and red colors indicate negative and positive $dCF_{T,RH}$, respectively. A white X indicates that the cell value is not significantly different from zero (Wilcoxon rank test, $p < 0.05$). **Note that each underlying Wilcoxon rank test has a 5% chance of yielding a false positive indication of statistical significance or an unknown (but likely much higher) chance of yielding a false negative result. Consequently, the distribution of Xs should not be over-interpreted. The number of Xs, however, provides an objective way to test whether the evidence for an effect on the grid as a whole is significant. This is consistently the case; in all panels, individually significant cells numbered more than expected at random (binomial test, $p < 0.001$)."**

*To further address the reviewer's comment, in the manuscript we have also better clarified the scientific reasons for including these cells. New text includes:*

"**Note that the $\overline{dCF_{T,RH}}$ value is based on all $dCF_{T,RH}$ data, including those from T and RH ranges where $dCF_{T,RH}$ is not significantly different from zero (i.e., as indicated by the white Xs in Figure 1). Including all data avoids biasing the results in favor of the meteorological conditions where $dCF_{T,RH}$ is most observable**."

*And in the Figure 4 caption:* "**In order to avoid obscuring emergent properties of the full dataset, the data [shown in Figure 4] include all meteorological conditions, including those where $dCF_{T,RH}$ are not significantly different from zero (as noted by a white X in Figure 1)**."

*Note that the cubic smoothing splines in Fig. 4 actually cross zero, so excluding those data that are not significantly different from zero would lead to less information on the system as a whole.*

23. Fig. 3: Does the significance indicated by the asterisk apply to both the relative percent changes and absolute changes? Please state in text.

*It does, and we now state this, as suggested.*

24. P7, L17, "1.7% to 0.7%": These values for sea ice do not match what I see, which is -2% to 1% (unless you maybe meant only at higher altitudes?).

*There was actually a negative sign in front of the 1.7% (it was easy to miss, being present on the line above, due to the formatting of the ACPD template). So rounded up, the -1.7% goes to -2%, and the 0.7% is rounded to 1%.*

25. P8, L1, "more influential at the lower temperatures": The values are near dCF=0 for the higher altitude points so is this an accurate statement, especially given the dramatic drop off over sea ice with lower altitude (warmer temperature) suggesting the dominance of the stability criterion?

*From the previous sentence, please note that we had been specifically discussing low altitude cases. To avoid confusion, we have re-worded as follows:*

"Also, $dCF_{T,RH}$ changes at high $dBC_{T,RH}$/low altitude are larger over sea ice (Fig. 4). This finding suggests that aerosol microphysical impacts on **low-altitude** clouds are more influential at the lower temperatures and/or more stable conditions over sea ice."

26. P8, L6, "(Fig. 3)" → "(Fig. 3a)" for clarity.

*Changed as suggested.*

27. P8, L11, "Over sea ice" → "From Table S2, over sea ice..." for clarity, as otherwise it was not clear what supported the last sentence in the paragraph.

*Changed as suggested.*

28. P8, L18, "where LPC fractions were highest": Source for statement?

*This sentence has been changed as follows:* "The relative fraction of liquid clouds was reduced between 0.6-1.5 km **(Fig. 3a)**, where LPC fractions were highest **(see blue bars in Fig. 3b)**."

29. P8, L23, "An analysis..." → "We analyze the difference in precipitation frequency; however, an analysis..." (otherwise, the reader knows what you will not do, but it has not been stated what will be done).

*Changed as suggested.*

30. P9, L1, "~91% of the MPCs": Shown where?

*The new text reads,* "Over sea ice, ~94% of MPCs were present below 4 km **(Fig. 5a)**."

31. P9, L31, "night, potentially" → "night. This potentially leads to..." (break up the long sentence that also contains opposing points of view). And, which point of view does your study support?

*Note that some new text was added based on a relevant paper that recently came out. The new text reads:*

"**Although BC itself is not thought to be a good source of INPs (Vergara-Temprado et al., 2018),** combustion aerosols associated with BC might act as ice nucleating particles (INPs) (Kanji et al., 2017) at the extreme cold temperatures found at high-altitude Arctic polar night. **This could** potentially lead to smaller, more numerous ice particles that precipitate less (Lohmann and Feichter, 2005)**, in line with our observations**, although some models suggest that INPs may instead lead to larger ice crystals in cirrus clouds compared to homogeneous freezing (Heymsfield et al., 2016)."

32. P11, L12: Physically, why would one expect a larger aerosol effect for greater atmospheric stability?

*The text has been modified as follows:*

"In general, aerosol microphysical effects were most observable where the highest aerosol effect would be expected: at lower altitudes where aerosol concentrations are often higher (Devasthale et al., 2011b) and over sea ice, where atmospheric stability is greater**, and aerosol microphysical effects on clouds are less likely to be overwhelmed by meteorological factors such as high vertical velocity**."

33. P11, L25: Can you give a "for example" about what other cloud property relationships might exist?

*The reviewer refers to the following text:*

"Furthermore, these observations leave open the possibility that other cloud property relationships with $dBC_{T,RH}$ exist, but are not observable with the available data."

*We are not sure we want to hypothesize in the paper, per se, but such relationships could include, for example, significant changes in cloud phase over open ocean that were just not observable in our 2-year dataset given the available sample sizes.*

34. Need the "author contributions" section for ACP

*These have now been added, thanks.*

35. L49-52: The concern about the variations in BC:OC ratios seems misplaced to me since the CALIPSO cannot tell the difference between those aerosol types.

*To clarify, this uncertainty is mentioned exactly because CALIPSO cannot tell the difference between those aerosol types. The concern is that if we assume that CALIPSO can detect combustion aerosols in general (including OC and BC aerosols), and we are using model BC as a proxy for all of these combustion aerosols, we may underestimate combustion aerosol layers with very high OC:BC ratios.*

36. L33-35, "it is unclear how thick an observed CALIPSO aerosol layer (measured in meters) must be to influence the average BC concentration in an altitude range...": The part "must be to influence the average BC concentration" seems odd in that there is no "influence" on the average BC concentrations. Please reword.

New text now reads: "As such, it is unclear how observed CALIPSO aerosol layer **thickness** (measured in meters) **would relate to** the average BC concentration in an altitude range equivalent to the FLEXPART model's vertical resolution (measured in kilometers)."

37. L38: "locations of false" → "locations of FLEXPART false"

*Changed as suggested.*

38. L63-64: Please explain a bit more about how the FLEXPART and CALIPSO data are compared. Specifically, the text refers to percentages of a "layer volume"; is that to say that the CALIPSO layering is converted into a binary present/not present mask and compared to the equivalent binary from FLEXPART? If so, is there a criterion used for the binary CALIPSO masking?

*To clarify, we have added the following text:*

"Vertical aerosol layer distribution was obtained from CALIPSO v. 4.10 level 2, 5-km merged aerosol and cloud layer data (CALIPSO Science Team, 2016) at 532 nm. These data are collected at 30-m vertical resolution up to 8.2 km, and at 75-m resolution between 8.2-8.5 km. Aerosol-containing profiles were required to be cloud-free and to have cloud-aerosol detection (CAD) scores > 70, indicating high confidence in cloud and aerosol separation. **For each clear-sky polar night profile during our sample period, we noted the fraction of each FLEXPART model vertical layer (0.6 to 1.5 km, 1.5 to 2.5 km, 2.5 to 4 km, 4 to 6 km, and 6 to 8.5 km) that was filled by an observed CALIPSO aerosol layer. From these fractions, ranging from 0 to 1, weighted averages were calculated on a horizontal basis at each altitude level over the Arctic Ocean region (see Figure S6).**"

"Based on the above assumptions, model false negative rates in clean conditions are likely to be highest when CALIPSO aerosol layers are observed in a large fraction of the model altitude layer. Average "clean" FLEXPART vertical layers often contain some CALIPSO-observed aerosol layers within them. **Based on a weighted-average grid analysis of data throughout the study period (Fig. S3a), CALIPSO aerosol layers are present in,** on average, ~19-27% of **FLEXPART** layer volumes. The actual BC concentrations of these aerosol layers are unclear."

39. L66: For clarity, recommend "so the fractions estimated" → "so the model false negative fractions estimated"

*Changed as suggested.*

40. L69: Remove "However". It implies a course change from the prior text but one is not present.

*Changed as suggested.*

41. L76: For clarity, "sea ice" → "sea ice (the pink line)"

*Changed as suggested.*

42. L81: For clarity, recommend "detect non-dilute aerosol" → "detect (non-dilute) aerosol"

*Changed as suggested.*

43. L82: "are likely to be"; shouldn't this be "will be" (?) since it is how you have defined false positives?

*Yes, this is correct, we have changed the text to clarify this.*

44. L86-87, "These aerosol...": I was looking for a figure to support the statement but it seems that one is not present? If there is, please indicate; if there isn't, please indicate "not shown."

*We now indicate Fig. S4.*

45. Figure S4: Please rescale the y-axis to cover the range of the bars plotted (i.e., most "polluted" bars exceed the plot range).

*Changed as suggested.*

46. Table S2: The values in the square brackets are defined in the caption, but what are the values in the parentheses that precede the square brackets? Recommend rearranging to be in column format as the font size to fit in portrait is too small to read easily.

*This Table is now Table S1. We have clarified that the round brackets are the bootstrapped 95% confidence intervals for the weighted mean, and have rearranged in column format, as recommended.*

---

## Author Response (AR2)

Comments to the Author:

Dear Authors,

I have reviewed your revised manuscript and acknowledge the effort that has gone into the important revisions. Before I accept it finally, I wish to make an observation that I would like you to consider prior to publication in ACP.

The paper's title speaks about the influence of BC aerosol on Arctic clouds. From the Abstract and Summary/Conclusions, it is clear that the meteorological variability explains most of the effect - in fact 91% of CF. After accounting for the meteorological variability, BC aerosol-related changes are 6% or less in the various parameters. This is an important message! It therefore seems appropriate to state this as your most important result rather than to focus on the small aerosol effect. I realise that everything I have said is stated in the Abstract, but it seems like you are trying to salvage an aerosol effect rather than emphasise the meteorological influence. The same is true of the opening sentences of the Summary/Conclusions.

I think your paper would be much stronger if this came out more clearly. This doesn't detract from the value of your study. In fact, the clarity would be welcome.

I'm not suggesting major changes - rather some rephrasing/shift in emphasis of your message. This may be appropriate in other parts of the paper as well.

On a minor point, on pg 2, you state: "reflect the true microphysical effects of combustion-derived aerosols over sea ice and open ocean regions more accurately".

I think "true" and "more accurately" are a bit of an overstatement. How about "are more representative of the microphysical effects of..."

Sincerely,
Graham Feingold

*Dear Dr. Feingold, Co-Editor,*

*Thank you for your suggestions to improve the paper! We have changed the wording you mentioned on page 2 as you suggested.*

*Regarding your other comments, you make a good point - we certainly don't want to de-emphasize the importance of the meteorology over that of the aerosols. We have made several changes in the manuscript to make this clearer (see new version), including stating more emphatically that meteorological co-variability drives a majority of the differences in cloud properties from clean conditions.*

*That said, based on the comments above, we want to clarify two key bits of information.*

*First, our conclusion is that BC aerosols could still explain up to 40% of the variability in CF differences from clean conditions. The 91% value you mention above is only an upper*

*limit on the possible influences from T and RH co-variability with BC based on our analysis. Our original intent of putting in the "up to 91%" statement in the abstract was to emphasize why it is so important to account for meteorological co-variability. However, we now realize that without better context, this statement could be over-interpreted. Therefore, we now put the full range of estimated signal from meteorological co-variability (57-91%) in the abstract, along with more complete information on how this range was derived in the methods. We will also note that aerosols can also account for up to 40% of the variance.*

*Secondly, BC-aerosol-related changes to cloud properties are small (< 6% from Fig. 3[1]), but so may be the effects of co-varying meteorology on total cloud properties. In this paper we didn't go into great detail in estimating those co-varying meteorology impacts. However, based on the GAM results, even if co-varying meteorology explained the maximum 91% of the variance in Table 1 dCF values (from the "difference" column in the "Mean CF" row), these dCF values are still small relative to average CF in the same Table.*

*So, in summary, in this paper we show that there are large uncertainties from co-varying meteorology. We agree it is worth emphasizing that co-varying meteorology is more important than the effects of the aerosols, and we have tried to do so in the paper. A strength of this paper is that despite high uncertainties from co-varying meteorology, we have devised a way to constrain the aerosol effects on cloud properties. However, we also need to be careful to avoid overstating our findings with regard to the importance of co-varying meteorology due to these fairly high uncertainties. We have tried to clarify these uncertainties better in the paper to avoid being misinterpreted.*

*Best regards, and thanks for your help!*

*-The Authors*
* * *
[1] As a reminder, absolute values for the weighted mean of $dCF_{T,RH}$ in Figure 3 indicate how much more or less cloudy the sky would be on average (e.g., 2% less cloudy). The relative values for the weighted mean of $dCF_{T,RH}$ basically indicate how much bigger or smaller average individual cloud extent would be because of changes in aerosols relative to clean conditions (e.g., 6% smaller).

[revised manuscript text omitted]